# GUIDANCE WATERMARKING FOR DIFFUSION MODELS

**Enoal Gesny**[1]**, Eva Giboulot**[1]**, Teddy Furon**[1] **& Vivien Chappelier**[2]
[1]Inria, Rennes, France
[2]LABEL4.AI, Rennes, France
`{enoal.gesny}@inria.fr`

## ABSTRACT

This paper introduces a novel watermarking method for diffusion models. It is based on guiding the diffusion process using the gradient computed from any off-the-shelf watermark decoder. The gradient computation encompasses different image augmentations, increasing robustness to attacks against which the decoder was not originally robust, without retraining or fine-tuning. Our method effectively convert any *post-hoc* watermarking scheme into an in-generation embedding along the diffusion process. We show that this approach is complementary to watermarking techniques modifying the variational autoencoder at the end of the diffusion process. We validate the methods on different diffusion models and detectors. The watermarking guidance does not significantly alter the generated image for a given seed and prompt, preserving both the diversity and quality of generation.

## 1 INTRODUCTION

Diffusion models have been the touchstone of the recent advancements in image generation. Once challenging tasks, such as text-to-image generation, image-to-image translation, super-resolution, or inpainting, are now performed with ease and flexibility. Various optimizations (Song et al., 2021; Rombach et al., 2022a; Dao, 2024) and the proliferation of accessible interfaces (Ramesh et al., 2022; Zhang et al., 2023; von Rütte et al., 2024) have made this technology accessible to users without technical know-how and high-end hardware. Generative AI now creates high-quality, diverse, and photorealistic images that are perceptually indistinguishable from real images.

Regulating entities have identified the risks posed by such technology (USA, 2023; China, 2023; Europe, 2023). Notably, there is an essential demand regarding the **identification and traceability of AI-generated content** (Fernandez et al., 2024b). Among existing solutions (such as metadata (C2PA, 2024) and forensics (Corvi et al., 2023)), digital watermarking stands out as a key technique.

Watermarking embeds imperceptible identifiers into images, making them detectable by private decoders. This mature technology has many applications, including copy protection, audience measurement, content identification and monetizing, broadcast monitoring (DWA). It has recently been adapted to the identification of generated content. Among many scenarios listed by the NSA (2025), one is to warn users of social networks or Internet search engines that these images are not real, another is to filter out AI-generated images from the training sets of future generative AIs to avoid a model collapse (Bohacek & Farid, 2025). In both cases, the watermark detector analyses billions of images. The requirement of utmost importance is a provably low false alarm rate, *i.e.* the probability of flagging a real image as AI-generated.

Numerous designs have been proposed for text (Kirchenbauer et al., 2023), voice (San Roman et al., 2024), and generated image (Fernandez et al., 2023). For this latter media, the strategy ranges from post-generation watermarking to clever modifications of the generation delivering content that is 'intrinsically' watermarked (Wen et al., 2023; Yang et al., 2024b; Huang et al., 2025; Fernandez et al., 2023). The first method is referred to as *post-hoc* and the second as *in-generation* watermarking.

This paper presents a principled methodology for converting any *post-hoc* watermarking into an *in-generation* scheme for any diffusion model. The idea is to guide the diffusion process towards generating images that are intrinsically deemed watermarked by any arbitrary watermark detector. We summarized the method in Figure 1. Our contributions are the following:

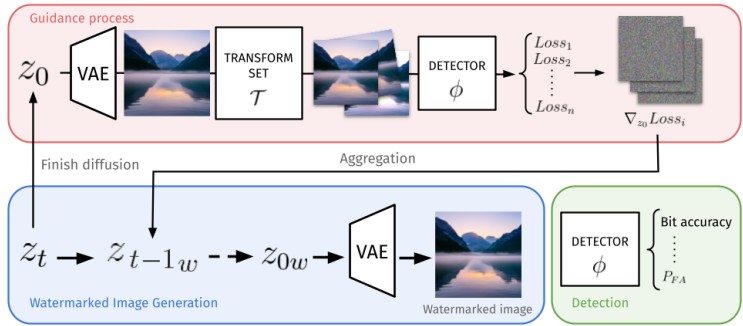

Figure 1: System-level diagram of the proposed guidance-based watermarking method.

1. Our method is the first to embed the watermark during the diffusion process itself with the use of guidance

2. It does not necessitate any retraining of the diffusion model.

3. It inherits from the robustness of the watermark detector, but can also improve it against new targeted attacks without retraining the detector.

4. It strikes a balance between complete modification of the semantic content (seed-based schemes) and the addition of an invisible signal (VAE-based and post-hoc schemes).

## 2 RELATED WORK

### 2.1 DIFFUSION MODELS

Diffusion has emerged as a powerful framework leveraging iterative denoising to generate high-quality images. Starting from a forward process gradually corrupting data with Gaussian noise:

$$q(z_T \mid z_0) = \prod_{t=1}^{T} q(z_t \mid z_{t-1}), \quad \text{with} \quad q(z_t \mid z_{t-1}) = \mathcal{N}(z_t; \sqrt{1-\beta_t}z_{t-1}, \beta_t I), \quad (1)$$

where $\beta_t$ controls the noise schedule, the Denoising Diffusion Probabilistic Model (DDPM (Ho et al., 2020)) learns a reverse process to iteratively denoise through parameterized transitions:

$$p_\theta(z_{t-1} \mid z_t) = \mathcal{N}(z_{t-1}; \mu_\theta(z_t, t), \Sigma_\theta(z_t, t)). \quad (2)$$

Denoising Diffusion Implicit Models (DDIM (Song et al., 2021)) extends this framework by introducing non-Markovian sampling, enabling deterministic generation through an ODE-like process:

$$z_{t-1} = \sqrt{\bar{\alpha}_{t-1}} \left( \frac{z_t - \sqrt{1-\bar{\alpha}_t}\,\epsilon_\theta(z_t, t)}{\sqrt{\bar{\alpha}_t}} \right) + \sqrt{1-\bar{\alpha}_{t-1}}\,\epsilon_\theta(z_t, t), \quad (3)$$

where $\bar{\alpha}_t = \prod_{s=1}^{t}(1-\beta_s)$. For the sake of simplicity, we omit the user prompt conditioning $\epsilon_\theta$. Subsequent advancements improve efficiency through latent space optimization balancing computational cost with perceptual quality (Dhariwal & Nichol, 2021; Nichol & Dhariwal, 2021). Modern image generators are Latent Diffusion Model (LDM) working in a latent space $\mathcal{Z}$. From the initial vector $z_T \in \mathcal{Z}$ drawn as a white Gaussian vector, the variational auto-encoder (VAE) transforms the final latent $z_0 \in \mathcal{Z}$ into an image $x_0 = \text{VAE}(z_0)$ in the image space $\mathcal{X}$.

### 2.2 GRADIENT-BASED GUIDANCE

Gradient-based guidance mechanisms in diffusion models enable precise control over generation by incorporating external signals through backpropagated gradients during the denoising. Introduced by Dhariwal & Nichol (2021), this approach modifies the sampling trajectory using auxiliary objectives, such as classifier scores or perceptual losses. For instance, Jeanneret et al. (2022) implement gradient-based guidance to steer the diffusion process toward generating counterfactual examples to

explain the prediction of a given classifier. Given a query image, the goal is to make the diffusion model generate an image as close as possible to the query but classified differently.

Our work is inspired by this trend, considering that a watermark decoder is indeed a classifier. Yet, we do not have a query image to start with, but a user prompt. We include an augmentation layer to gain robustness, a concept irrelevant for counterfactual examples.

## 2.3 WATERMARKING IMAGES GENERATED BY DIFFUSION MODELS

**Post-hoc** Traditional image watermarking embeds a watermark signal into an original image (Cox, 2008). In zero-bit watermarking, the detector decides whether the watermark is present or absent, while in multi-bit watermarking, the decoder retrieves the hidden binary message from the image under scrutiny. Recent advancements leverage the capabilities of a pair of deep neural networks to embed and detect/decode the watermark: The foundational HiDDeN framework of Zhu et al. (2018) established such an end-to-end pipeline inspiring subsequent adaptations such as TrustMark (Bui et al., 2023), VideoSeal (Fernandez et al., 2024a), or InvisiMark (Xu et al., 2025). The training minimizes a loss combining a perceptual distance between the original and watermarked images with a multi- or zero-bit classification loss. An augmentation layer distorts the watermarked image before passing it to the detector/decoder in order to improve the robustness.

Traditional watermarking is a communication channel through a host content, whose theoretical pillar is based on the work of Costa (1983) establishing the capacity of a side-informed communication scheme. Its main message is that the original image should not be seen as a source of noise limiting the capacity of the hidden communication channel, but as a side-information known while emitting the watermark signal. Yet, it is difficult in practice to be sure that the host image is not interfering with the watermark; especially in zero-bit watermarking (Comesana et al., 2010; Furon, 2017).

Post-hoc means that the generated image is the original image forwarded to a traditional watermarking scheme before being returned to the user. The main weakness is that it is not specific to generative AI. Although these methods demonstrate progress in robustness, they operate as external add-ons rather than integral components of the generative process.

**In-generation** Stable-Signature pioneered the in-generation approach by merging the final step of the Stable Diffusion model, *i.e.* the VAE, with a post-hoc watermark embedding (Fernandez et al., 2023). To do so, it fine-tunes the VAE using a loss combining a perceptual distance between the images generated by the new and the original VAE together with a loss on the decoded message when the generated image goes through a given pre-trained watermark decoder.

In stark contrast, Wen et al. (2023) claim that there is no such thing as an original image in GenAI watermarking. The user will never see the image generated without a watermark. The model is sampling images not related to any reference image; therefore, controlling the distortion introduced by the watermark, like in post-hoc watermarking or Stable Signature, is a meaningless constraint.

A second difference is that Wen et al. (2023) embed the watermark signal before the diffusion process: Tree-Rings crafts a seed $z_T$ in the latent space with a secret pattern. A third difference is that Tree-Rings first defines the way to sample watermarked images, and then designs a possible detector. From an image under scrutiny, the detector first estimates the seed by inverting the diffusion process and computes the distance to the secret pattern. The image is deemed watermarked if this distance is below a given threshold. It offers fair robustness against geometric attacks by enforcing some structure in the secret pattern. Yet, our appendix C shows that the false alarm rate is not under control.

Yang et al. (2024b) improve this idea in several aspects: First, Gaussian Shading takes care of crafting seeds following a Gaussian distribution as required by many diffusion models. It is a multi-bit watermarking with excellent robustness against valuemetric attacks thanks to a repetition error correcting code. However, it is not robust to geometric attacks.

Huang et al. (2025) notice that the semantic content of the generated image changes with the strength of the Tree-Ring watermark. Their proposal, RoBIN, postponed the watermark embedding to an intermediate step within the diffusion process. This makes a compromise between maintaining the semantic of the image (unlike Tree-Ring) while not caring about the norm of the additive watermark

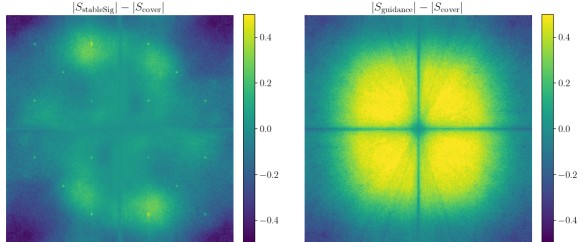

Figure 2: Differences of (log)-spectrum of generated images with and without watermarking. Left: Embedding by the VAE (Stable Signature Fernandez et al. (2023)). Right: Embedding during the diffusion (Ours). Appendix A details the computation of these spectrums. Stable-Signature embeds a signal in a specific frequency band, whereas our method distributes the signal over the full spectrum.

signal (unlike post-hoc and Stable Signature). The main problem is that, similarly to Tree-Ring, the false positive rate is high and does not come with a theoretical guarantee.

## 3 MOTIVATIONS

We borrow from Stable Signature the idea that the decoders of traditional watermarking schemes are quite robust thanks to the augmentation layer considered during their training. Moreover, some designs take great care of controlling the false alarm rate (see, for instance, (Fernandez et al., 2023, Fig. 12)). Therefore, these are good and sound starting points.

However, Stable Signature requires fine-tuning the VAE, which acts like an advanced upscaling: It upscales the latent representation to a large image and adds high-frequency details. Therefore, this in-generation watermarking technique focuses the watermark power on the high-frequency details. Figure 2 illustrates this fact on the left. The spectrum difference with and without Stable Signature watermarking shows the watermark energy spread in high frequencies. This explains the relatively low robustness of Stable Signature against low-pass filtering processes like JPEG compression. In contrast, our technique spreads the energy of the watermark all over the spectrum.

We also borrow from in-generation schemes the idea that watermarking should not be seen as the addition of a low-amplitude signal over an original image (Wen et al., 2023; Yang et al., 2024b). As such, PSNR is not an appropriate metric for GenAI watermarking. Yet, we agree with Huang et al. (2025) that the semantics of the generated image should not fluctuate due to the watermark.

In a nutshell, our goal is to sample images deemed as watermarked by a pre-trained detector. This conditioning of the sampling is made without any reference to an original image and as early as possible to plant the watermark in the semantic of the generated image.

## 4 OUR METHOD GUIDES THE DIFFUSION TO EMBED A WATERMARK

### 4.1 ASSUMPTIONS

The image generator is a Latent Diffusion Model defined by a latent space $\mathcal{Z}$, a number of diffusion steps $T$ with an associated scheduling $(\alpha_t)_{t \in [\![T]\!]}$ with $[\![T]\!] = \{1, \ldots, T\}$, a noise estimate model $\epsilon_\theta : \mathcal{Z} \times [\![T]\!] \to \mathcal{Z}$, and the function $\text{VAE} : \mathcal{Z} \to \mathcal{X}$ converting latent vectors to images. The diffusion generates an image $x_0$ from a seed $z_T$ through the following abstract update process:

$$\forall t \in [\![T]\!], z_{t-1} = \text{Diffusion}\left(z_t, \epsilon_\theta, t\right), \quad x_0 = \text{VAE}\left(z_0\right). \tag{4}$$

We keep the diffusion update mechanism Diffusion abstract since our method does not depend on the specific choice of solver for the diffusion process: It estimates the noise of a latent $z_t$ using $\epsilon_\theta$ at timestep $t$, outputting a denoised latent $z_{t-1}$ from this estimated noise.

The pre-trained watermark decoder/detector uses the extraction function $\phi : \mathcal{X} \to \mathbb{R}^M$ to compute $M$ raw logits. This function is a deep neural network easily differentiable thanks to backpropagation. For decoding, the decoded bits are the sign of the logits: $\hat{m} = \text{sign}(\phi(x))$, element-wise (Bui et al.,

2023; Fernandez et al., 2024a; Xu et al., 2025). $M$ is thus the watermark length. From a binary message $m \in \{0, 1\}^M$, the antipodal modulation outputs the vector $u_m = -(-1)^m$ component-wise, in $\mathbb{R}^M$. For detection, the image $x$ is deemed watermarked if the cosine similarity $\cos(\phi(x), u_m)$ is above a threshold, with $u_m \in \mathbb{R}^M$ a reference secret vector as in Fernandez et al. (2022).

A crucial assumption is that the extraction function provides a random feature $\phi(X)$ with an isotropic distribution in $\mathbb{R}^M$ when applied on a random non-watermarked image $X$, be it synthetic or real. This is approximately the case in Stable Signature as Fernandez et al. (2023) whiten $\phi(X)$ with a PCA.

## 4.2 GUIDED-DIFFUSION FOR WATERMARKING

Our method resorts to conditional sampling as introduced by Dhariwal & Nichol (2021) for DDIM and extended to other solvers by Lu et al. (2022). The differentiable detector $\phi$ along with a differentiable loss function $L : \mathcal{Z} \to \mathbb{R}_+$ guides the diffusion process. At each iteration, the estimated noise is modified by incorporating information from the gradient of the loss:

$$\hat{\epsilon}(z_t, t) := \epsilon_\theta(z_t, t) - \omega\sqrt{1 - \bar{\alpha}_t}\nabla_{z_t} \log L(z_t) \tag{5}$$

where $\omega$ is a scalar denoting the strength of the watermark guidance. This parameter must be carefully calibrated to ensure sufficient watermark detectability while maintaining image quality. The diffusion update is effectively replaced by $z_{t-1} = \text{Diffusion}(z_t, \hat{\epsilon}, t)$.

## 4.3 CHOICE OF THE LOSS FUNCTION

The loss function defined above is symbolic. In practice, it should depend on the message to be hidden (multi-bit) or the secret vector $u_m$ (zero-bit) and on the vector extracted from the image generated from the latent $z_t$. In other words, from a latent $z_t$, to gain access to the loss and its gradient, we complete the diffusion process from $t$ to $0$ and use the VAE, before applying the decoder/detector to the resulting image $x_0$, which we loosely denote as $x_0(z_t)$. To unify decoding (multi-bit) and detection (zero-bit), we propose the loss function $L : \mathcal{Z} \times \mathbb{R}^M \to \mathbb{R}_+$

$$L(z_t, u_m) := 1 - \frac{u_m^\top \phi(x_o(z_t))}{\sqrt{N}\|\phi(x_o(z_t))\|_2} = 1 - \cos(\phi(x_o(z_t)), u_m). \tag{6}$$

Our goal is to minimize the angle $\theta$ between $u_m$ and $\phi(x_o(z_t))$. In multi-bit watermarking, the decoding is exact if this loss is lower than $1 - \sqrt{1 - M^{-1}}$.

In zero-bit watermarking, this loss can be related to the following quantity, known as the $p$-value in statistics under some assumptions detailed in App. C:

$$p = \frac{1}{2}\left(1 \pm I_{\cos^2 \theta}\left(\frac{1}{2}, \frac{(M-1)}{2}\right)\right), \tag{7}$$

with $\cos(\theta) := 1 - L(z_t, u_m)$ and $I_x(a, b)$ is the regularized incomplete beta function. The sign is positive if $\cos(\theta) > 0$, negative otherwise. If a probability of false alarm $P_{\text{FA}}$ is required, the watermark is detected if the computed $p$-value is lower: $p < P_{\text{FA}}$. Hence, minimizing the loss amounts to minimizing the $p$-value for a watermarked image, which in turn increases the probability to be correctly detected.

## 4.4 ROBUSTNESS AGAINST IMAGE TRANSFORMATIONS

Until now, we controlled the diffusion to minimize the decoding loss for the untouched generated image $x_0$. A first enhancement minimizes the loss for an image modified with a chosen set $\mathcal{T}$ of image transformations $T : \mathcal{X} \to \mathcal{X}$, a.k.a. augmentations, ensuring a robust watermark. At each diffusion step, we compute the loss for an individual transformation $T$ redefined as $L(z_t, u_m; T) := 1 - \cos(\phi(T(x_o(z_t))), u_m)$. We compute the gradient for each new loss and aggregate them:

$$\hat{\epsilon}_\mathcal{T}(z_t) := \epsilon_\theta(z_t) - \sqrt{1 - \bar{\alpha}_t}\text{Agg}\left(\{\nabla_{z_t} \log L(z_t, u_m; T) \mid T \in \mathcal{T}\}\right). \tag{8}$$

The choice of aggregator Agg is crucial. The gradient directions might not agree for different transformations, leading to subpar performance if using a simple averaging. There exists an extensive

literature addressing this problem in multi-task learning (Liu et al., 2021) and byzantine federated learning (Guerraoui et al., 2024). We settled on the well-known PCGrad algorithm (Yu et al., 2020).

One advantage of this approach is that $\mathcal{T}$ can contain transformations for which the original feature extractor $\phi$ is not inherently robust. Section 5 shows this enhances the robustness of our method against these transformations without the need to retrain the watermark detector.

### 4.5 FAST AND CONTROLLED GUIDANCE FOR WATERMARKING

Our method is too computationally expensive, requiring $T(T+1)/2$ diffusion and gradient propagation steps. This section suggests two simplifications. First, we turn on the watermarking guidance at a step $T_w$, $0 < T_w < T$. Second, we simplify the gradient propagation along the backward diffusion by an identity transform. In other words, $\nabla_{z_0}$ replaces $\nabla_{z_t}$ in equation 8.

Finding a suitable guidance strength is cumbersome as it depends on the watermark decoder $\phi$ and the image generator. We propose to clip the gradient norm in order to control the amount of watermark signal added at each diffusion step:

$$\hat{\epsilon}(z_t, t) := \epsilon_\theta(z_t, t) - \omega\sqrt{1 - \bar{\alpha}_t}\frac{g}{\max(\eta, \|g\|)}, \quad \text{with } g = \text{clip}_\tau(\nabla_{z_0}\log L(z_t)) \tag{9}$$

with $\eta$ and $\tau$ to be chosen by the user – see Appendix B.

## 5 EXPERIMENTAL RESULTS

### 5.1 EVALUATION SETTING AND METRICS

**Diffusion models** We evaluate our method on three open-source diffusion models: *Stable Diffusion 2* (Rombach et al., 2022a), *Flux-1.0 dev* (Black Forest Labs, 2024), and *Sana* (Xie et al., 2024). We use their implementation available on HuggingFace. Of note, SD2 uses the *EulerDiscreteScheduler* solver, whereas Sana and Flux use the *FlowMatchEulerDiscreteScheduler*. This outlines that our method is agnostic to the diffusion mechanism. The images are generated from 1,000 prompts from the *Gustavosta/Stable-Diffusion-Prompts*[1] a series of prompts extracted from generated images which are meant to reflect more closely prompts used in a real environment. In Appendix D.2, we also report our experiments for 200 captions from the COCO dataset (Lin et al., 2014). Image size is set to $512 \times 512$, except for Flux, for which we chose $256 \times 256$ due to computation constraints.

**Watermarking detectors** We use the detectors from two state-of-the-art methods: Stable Signature (`SSig`, Fernandez et al. (2023)), and VideoSeal (`VS`, Fernandez et al. (2024a)). Their watermark lengths $M$ equal 48 (`SSig`) and and 256 (`VS`). We chose these schemes because they can be used as multi-bit decoders or zero-bit detectors (See Sect. 4.1). Appendix C experimentally verifies that the returned $p$-value is valid.

**Baselines** We consider the in-generation schemes Tree-Ring (Wen et al., 2023) for zero-bit detection, and Gaussian Shading (Yang et al., 2024b) for multi-bit decoding (256 bits). We set the maximum ring diameter of Tree-Rings to 18 and 10 for SD2 and Sana respectively and use 3 and 10 latent channels for the embedding. Appendix C shows that the $p$-value computed by the original implementation of Tree-Rings is incorrect, and describes some corrections to get reliable $p$-values. Unfortunately, we did not succeed to fix RoBIN Huang et al. (2025) detector so we exclude it from our benchmark. We also compare with state-of-the-art post-hoc watermarking schemes `VS` as well as the in-generation strategy of `SSig` fine-tuning the VAE.

**Our embedding** We denote by `G-VS` and `G-SSig` our watermark embedding guiding the diffusion with the decoders above. The augmentations used in the gradient computation are: Identity, JPEG compression with QF 50 and 80, brightness $+0.2$, contrast $\times 2$, and central crop $50\%$. These are augmentations used at the training of the decoders, our guidance is thus aligned with their robustness. The watermark guidance parameters are found by a grid search to provide the best trade-off between watermark detectability and image quality. Appendix B details their values.

---

[1]Available on Huggingface at `https://huggingface.co/datasets/Gustavosta/Stable-Diffusion-Prompts`. We filtered NSFW prompts for this work.

| LDM | WM | FID (↓) | CLIP (↑) | PSNR (↑) | LPIPS (↓) | Capacity(↑) | $P_D$ @ $10^{-10}$ $P_{FA}$ | $-\log_{10}(P_{FA})$ @ $P_D = 0.9$ |
|---|---|---|---|---|---|---|---|---|
| SD2 | | 5.0 | 0.330 | | | | | |
| SD2 | G-SSig | 2.3 | 0.332 | 19.6 | 0.22 | **27.7 (+19.3)** | **0.99 (+0.5)** | **16.3 (+12.2)** |
| SD2 | G-VS | 2.2 | 0.332 | 18.5 | 0.28 | **212.2 (+37.7)** | **1.0 (+0.0)** | **105.6 (+61.8)** |
| Flux | | 9.5 | 0.271 | | | | | |
| Flux | G-SSig | 9.3 | 0.271 | 25.4 | 0.07 | **26.6 (+16.6)** | **0.99 (+0.46)** | **16.6 (+12.8)** |
| Flux | G-VS | 9.3 | 0.269 | 26.0 | 0.07 | **192.5 (+16.0)** | **1.0 (+0.0)** | **72.8 (+24.3)** |
| Sana | | 4.3 | 0.346 | | | | | |
| Sana | G-SSig | 4.2 | 0.347 | 28.6 | 0.02 | **26.5 (+17.0)** | **0.98 (+0.41)** | **15.5 (+10.6)** |
| Sana | G-VS | 4.1 | 0.346 | 23.5 | 0.07 | **207.5 (+28.8)** | **1.0 (+0.0)** | **96.4 (+49.2)** |

Table 1: Comparison of image quality for several diffusion models and robustness metrics for our G-SSig and G-VS. In parenthesis, difference with their siblings SSig and VS. The robustness metrics are average over a set of attacks: Identity, JPEG compression with QF 50 and 80, brightness +0.2, contrast ×2, and central crop 50%.

**Quality**   The quality of the generation is gauged by the CLIP score between prompts and images (Hessel et al., 2021), and the Fréchet Inception Distance (FID) (Heusel et al., 2017) between generated images and 5,000 images from the COCO dataset. The watermark should not spoil these metrics. We also provide the PSNR and LPIPS score (Zhang et al., 2018) between the images generated with and without watermark, although these metrics are not suitable for generative AI (Sec. 2.3).

**Robustness**   or multi-bit decoding, the robustness is measured via the Random Coding Union (RCU) bound (see Eq.(162) in (Polyanskiy et al., 2010)). We embed random binary messages and measure the Bit Error Rate $\rho$ at the decoding side. Assuming a Binary Symmetric Channel with crossover probability $\rho$, the RCU is a lower bound on the maximum number of bits that can be reliably transmitted for a given watermark length $M$ and a word decoding error probability $\epsilon$ set to $10^{-3}$. This allows for a fair comparison of decoders with different watermark lengths. For zero-bit detection, the robustness is measured by the detectability $P_D$ of the watermark at extremely low $P_{FA}$. We expose our method to reach this regime without many samples in Appendix C. For completeness we also report (log)-ROC for each method and model. The following figures and tables are extracted from the full body of experimental results given in App. D.1.

## 5.2   Impact on the quality of the image generation

The semantic and image composition with or without a watermark are very close for a suitable guidance strength. The visual differences are slightly different tone, colors, or shape (see Fig. 3). This is different from the noise-like watermark signal of post-hoc or SSig and also from the drastic change of composition of Tree-Ring (Wen et al., 2023, Fig. 2). Yet, too much guidance strength leads to artefacts as depicted in App. B.

The left part of Table 1 provides the quantitative assessment of the quality of generated images. As expected, the PSNR is relatively low whereas no differences are observed with respect to the FID and CLIP score. This confirms that the watermarked images are qualitatively similar to the non-watermarked images despite local differences for the same prompt and seed.

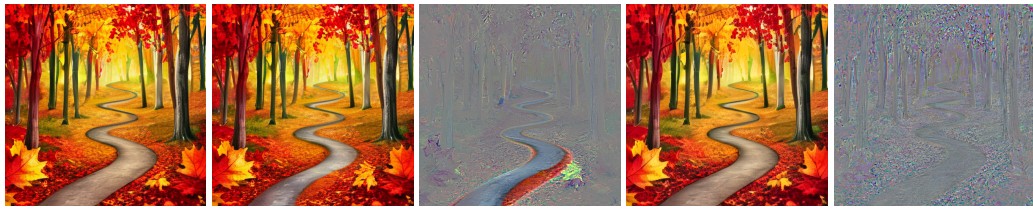

Figure 3: From left to right: 'A vibrant autumn forest with red, orange, and yellow leaves and a winding path' generated by Sana (1) without watermarking, (2) watermarked with our G-SSig, (3) difference (2)-(1) , (4) watermarked with Stable Signature SSig, (5) difference (4)-(1)

## 5.3 Comparison with in-generation schemes

Table 2 compares our performances with two in-generation schemes: Gaussian Shading for multi-bit decoding and Tree-Rings for zero-bit detection. By design, Gaussian Shading is quite robust to valuemetric attacks (an image processing which perturbs the pixel values) but absolutely not robust to geometric attacks (like shift, crop, rotation, . . . ). As for our method, we inherit from the robustness of the pre-trained decoder. For instance, both `G-VS` and `G-SSig` are robust to such a strong crop because the decoder saw it during its training. This allows our method to substantially surpass the performance of other in-generation schemes. The same holds for zero-bit detection. By design Tree-Rings is robust against rotation but not crop.

| WM scheme | Identity | Contrast (x2) | JPEG (Q=50) | Gaussian Blur 3 | Rotation 90 | Crop 50% |
|---|---|---|---|---|---|---|
| multi-bit decoding (capacity in bits) | | | | | | |
| Gaussian Shading | 221 | 211 | 181 | 216 | 0 | 0 |
| G-SSig | 32 | 29 | 24 | 25 | 27 | 21 |
| G-VS | **222** | **219** | **197** | **220** | **194** | **206** |
| zero-bit detection ($-log_{10}(P_{FA})$ @ $P_D = 0.9$) | | | | | | |
| Tree-Rings | 11.7 | 6.5 | 4.3 | 9.1 | 0.8 | 0.4 |
| G-SSig | 21.9 | 19.8 | 14.7 | 16.6 | 17.4 | 11.4 |
| G-VS | **154.6** | **130.6** | **89.2** | **150.3** | **100.7** | **101.9** |

Table 2: Comparison with two in-generation schemes for Stable Diffusion v2.

**Adversarial Attacks** Adversarial attacks have demonstrated strong effectiveness against traditional in-generation, seed-based watermarking schemes. Figure 4 shows the results of three representative attacks, VAE purification (Liu et al., 2020; Abdelnabi & Fritz, 2021), regeneration (Nie et al., 2022), and the average attack (Yang et al., 2024a), applied to respectively 500, 500 and 50 images generated with `G-VS`, Tree-Rings, and Gaussian-Shading.

We perform VAE purification by applying in succession the encoding and decoding function of the original VAE of the diffusion model used to generate the image. The regeneration attack is performed by encoding the watermarking image, adding noise to the latent corresponding to a given step and finishing the diffusion process. For each image, we use 2,4,6,8 and 10 steps. The average attack is performed under a gray-box setting: for each watermarking scheme and a **fixed key**, we compute the average residual between a set of watermarked and non-watermarked images. Our guidance-based method is content-dependent as shown in Appendix D.3, unlike Gaussian-Shading and Tree-Rings. Consequently, the average residual for the guidance-based approach has lower amplitude than for the other two methods. To ensure a fair comparison, we enforce a constant PSNR across methods by normalizing the attacking signal. All other attacks are performed over images with variable keys.

Figure 4 shows that `G-VS` is always more robust than Tree-Rings, and particularly for the average attack. It also demonstrate empirically that Gaussian-Shading is a little more robust against regeneration and VAE purification due to the redundancy of the signal, but not at all against the average attack whereas `G-VS` is robust against the three studied attacks.

## 5.4 Comparison with post-hoc watermark embedding and SSig

**Multi-bit performance** The right part of Table 1 shows the robustness metric averaged over the following benchmark attacks: Identity, JPEG compression with QF 50 and 80, brightness +0.2, contrast ×2, and central crop 50%.

**Zero-bit performance** we report the detectability at a low $P_{FA} = 10^{-10}$. In this regime, we double the performance of `SSig`. Since `VS` is already perfectly detectable in this regime, we provide a more fine-grained analysis in Figure 5. The (log)-ROC curves demonstrate a large gap in performance in favor of the guidance methods whatever the model and the method: for any arbitrary $P_{FA}$, the detectability is always significantly higher. For instance, an absolute gain between 20% and 10% is always observed for `G-VS` compared to `VS` in the low $P_{FA}$ regimes. The case for `SSig` is even more clear cut: `G-VS` stays perfectly detectable even at $P_{FA}$ where `SSig` has zero detectability.

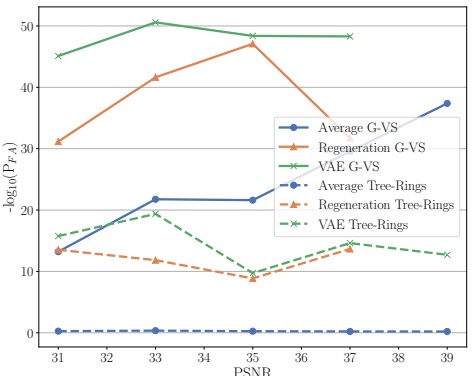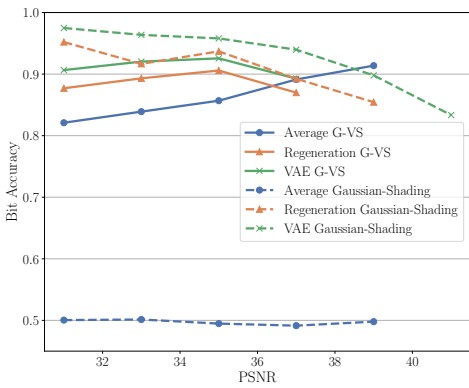

Figure 4: Adversarial attacks applied to Sana images watermarked with Tree-Rings, Gaussian-Shading or `G-VS`. Regeneration attacks and VAE purification are computed over 500 images while the average attack is computed over 50 images. The mean $-log_{10}(P_{FA})$ @ $P_D = 0.9$ and the mean bit accuracy are reported respectively for Tree-Rings and Gaussian-Shading.

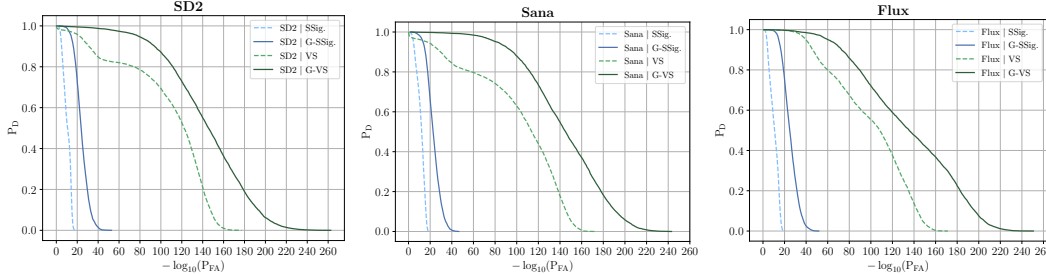

Figure 5: Probability of detection $P_D$ of post-hoc and corresponding guided methods as a function of the $P_{FA}$ for different models. The curve is shown over all studied augmentations, with 1000 images generated form the *Gustavosta/Stable-Diffusion-Prompts* prompts for each augmentation.

**Unknown augmentations** So far, the watermarking schemes were benchmarked against known attacks, *i.e.* attacks used as augmentations during the training of the decoder. We investigates whether our method can improve the robustness against unknown attacks by encompassing them in the gradient computations. This avoids retraining a decoder with a new set of augmentations. It happens that `SSig` is not robust against a 90-degree rotation or a median filtering. Figure 6 shows that we drastically improve the performance by encompassing these attacks. In the original work, Stable-Signature was shown to be easily removable by passing the image through the original VAE – a simple purification attack. We show in Figure 7 that our method is robust to such attacks *"for free"*. Indeed, since the gradient has to be back-propagated through the (unwatermarked) VAE, such attacks are *implicitly* within the transform set.

## 5.5 COMPUTATION COST

As hinted in Section 4.5, it is possible to apply guidance only during the last diffusion steps while remaining effective. The table 3 reports the performance of guided diffusion when applied for different numbers of steps. We report results with `VS` , as it is the best baseline detector, but we focus on comparing the computation cost with other in-gen watermarking methods. Overall, comparable performance to post-hoc methods can be achieved with 15 guidance steps, whereas 10 and 5 guidance steps are sufficient to outperform Gaussian-Shading and Tree-Rings respectively. It should be noted that seed-based methods require an inversion of the diffusion process for detection, increasing its cost compared to post-hoc methods. For both Gaussian-Shading and Tree-Rings, only 4 inverse diffusion steps are required for good detection on Sana and Flux. However SD2 requires 50 steps. Unlike these methods, ours does not require extra steps for decoding making it up to 50 times faster at detection time.

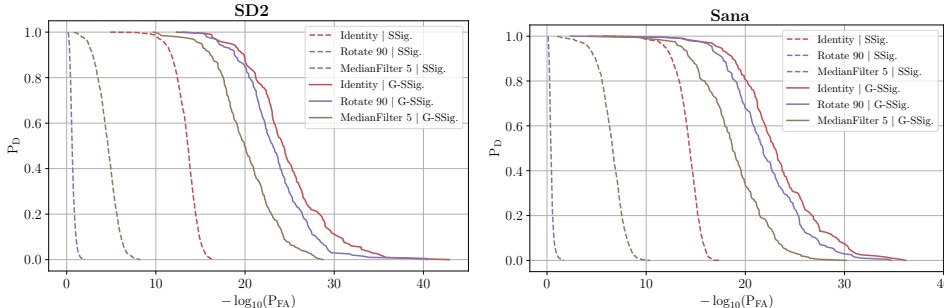

Figure 6: Zero-bit detection on Stable-Signature with SD2 and Sana. The guidance patches a weakness of the decoder by encompassing an unknown attack in the gradient computation. The curves were computed over 200 images from *Gustavosta/Stable-Diffusion-Prompts* for each attack.

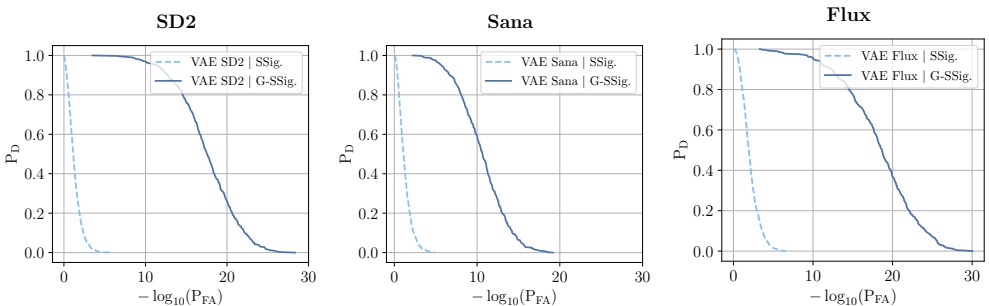

Figure 7: Probability of detection $P_D$ of post-hoc and corresponding guided methods as a function of the $P_{FA}$ for different models, under the attack using the original VAE to remove the watermark. The curves were computed over 200 images from *Gustavosta/Stable-Diffusion-Prompts* for each attack.

| WM scheme | | Capacity($\uparrow$) | $-log_{10}(P_{FA})$ @ $P_D = 0.9(\uparrow)$ | Steps($\downarrow$) |
|---|---|---|---|---|
| G-VS | | **207.5** | **96.4** | 325 |
| G-VS last 15 | | **178.0** | **70.0** | 130 |
| G-VS last 10 | | **117.7** | **36.9** | 70 |
| G-VS last 5 | | 15.3 | 6.2 | 35 |
| VS (post-hoc) | | 178.7 | 47.2 | 25 |
| Tree-Rings (in-gen) | | – | 0.70*/11.0 | 25 +det |
| Gaussian Shading (in-gen) | | **119.0** | 0.37*/28.5 | 25 +det |

Table 3: Comparison of the performances for several robustness metrics depending on the number of guidance step with Sana. The number of diffusion steps assume standard generation for VS. The parameters are the same used in Table 1 referenced in Table 4. Steps refers to the number of diffusion steps. *+det* indicates the additional diffusion steps required for detection. Since Gaussian-Shading and Tree-Rings are **not** robust to crop, we provide the $P_{FA}$ @ $P_D$ with (left) and without (right) the cropping augmentation.

## 6 CONCLUSION AND LIMITATIONS

This work introduces a new watermark embedding for latent diffusion models converting any *post-hoc* watermarking to *in-generation* without retraining of the model. Our method inherits the robustness from the baseline and can also improve it against attacks never seen by the decoder.

Limitations include a robustness that depends on the visual content of the image to be generated, and a generation time that needs 2 to 13 more steps ; however the decoding time is 50 times faster compared to other in-generation schemes such as Tree-Ring (Wen et al., 2023) and Gaussian Shading (Yang et al., 2024b).

## REPRODUCTIBILITY STATEMENT

All models, datasets, and detectors used in this work are detailed in Appendix G. `https://github.com/EnoalG/Guidance-Watermarking-for-Diffusion-Models` is the repository of the official implementation. The implementation relies on PyTorch and the diffusers library, with fixed random seeds. All hyperparameters used in the experiments are listed in Table 4, while other choices are specified at the beginning of Section 5.

## ACKNOWLEDGMENTS

Experiments presented in this paper were carried out using the Grid'5000 tested, supported by a scientific interest group hosted by Inria and including CNRS, RENATER and several Universities as well as other organizations (see `https://www.grid5000.fr`). This research is part of the PEPR Cyber Compromis project, ANR-22-PECY-0011.

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

## A  COMPUTATION OF THE SPECTRUM IN FIG. 2

From a batch of $N$ RGB images $\{x^{(i)}\}_{i=1}^{N}$ whose size is $L \times L \times 3$, we compute their 2D FFT representations channel-wise: $X^{(i)}(:,:,j) = \text{FFT}(x^{(i)}(:,:,j))$, $j \in \{1, 2, 3\}$. The spectrum $S \in \mathbb{R}^{L \times L}$ is computed as follows:

$$S(\ell, c) = \log\left(\frac{1}{3N}\sum_{i=1}^{N}\sum_{j=1}^{3}|X^{(i)}(\ell, c, j)|\right), \quad \forall(\ell, c) \in [\![L]\!]^2. \tag{10}$$

This spectrum is computed for the following batches of images generated with the same prompts: generated images without watermark ($S_{\text{cover}}$), generated images with Stable Signature ($S_{\text{StableSig}}$), generated images with guided watermarking for the decoder of Stable Signature ($S_{\text{guidance}}$). Figure 2 displays on the left the difference $S_{\text{StableSig}} - S_{\text{cover}}$, and on the right side $S_{\text{guidance}} - S_{\text{cover}}$.

## B  ABLATION STUDIES FOR HYPERPARAMETERS AND SIMPLIFICATIONS

### B.1  HYPERPARAMETERS

The algorithm relies on three hyperparameters to control the guidance. After computing the gradient, we clip a fraction of the most extreme values to limit their influence on the diffusion process. Similarly, if the gradient norm exceeds a threshold, we rescale it to that threshold. The parameter $\omega$ is a scalar denoting the strength of the watermark guidance 5. The values of these hyperparameters are reported in Table 4.

The $\omega$ parameter must be chosen carefully to balance detectability and image quality. Figure 8 illustrates that artifacts are generated when the $\omega$ value is exaggerated. Our choice sets this parameter to smaller values given in Table 4, used hereafter for the experimental assessment of the robustness. Figures 18, 19, and 20 show images generated with this reasonable choice of $\omega$ values. Figure 9 illustrates the influence of the parameters $\omega$ and $\tau$. Increasing $\omega$ improves bit accuracy but also amplifies the deviation from the non-watermarked image, as reflected by a lower PSNR. The parameter $\tau$ clips extreme values to prevent visual artifacts, increasing $\tau$ results in a higher PSNR with only a minor reduction in bit accuracy.

### B.2  IMPACT OF THE SIMPLIFICATIONS

Let Simplification 1 refer to the reduction in the number of diffusion steps used, as presented in Table 3. Table 5 complements these results by reporting the corresponding image quality metrics.

Let Simplification 2 denote the replacement of $\nabla_{z_0}$ by $\nabla_{z_t}$ in Eq. equation 8. The purpose of this simplification is to reduce the computational cost of the method. Table 6 reports results obtained on 200 Sana images watermarked with G-VS, both with and without simplification 1. The generated images exhibit lower quality metrics because the signal injected at each diffusion step is less stable than in the simplified version, due to full backpropagation through the diffusion process. Detection metrics are also reduced. Owing to the instability of the injected signal, the overall semantics of the image is altered. As shown in Appendix D.3, guidance watermarking is content-dependent; therefore, when global semantics drift, the watermark signal added at each diffusion step no longer follows the original diffusion trajectory. As a result, parts of the signal are progressively erased during the diffusion process, reducing final detectability.

| LDM | ‖ | % Clip ($\tau$) | Max norm ($\eta$) | $\omega$ |
|---|---|---|---|---|
| Stable-Diffusion 2 | ‖ | 10% | 0.3 | 250 |
| Flux | | 10% | 0.5 | 500 |
| Sana | ‖ | 10% | 0.5 | 600 |

Table 4: Values of the hyperparameters used for each model.

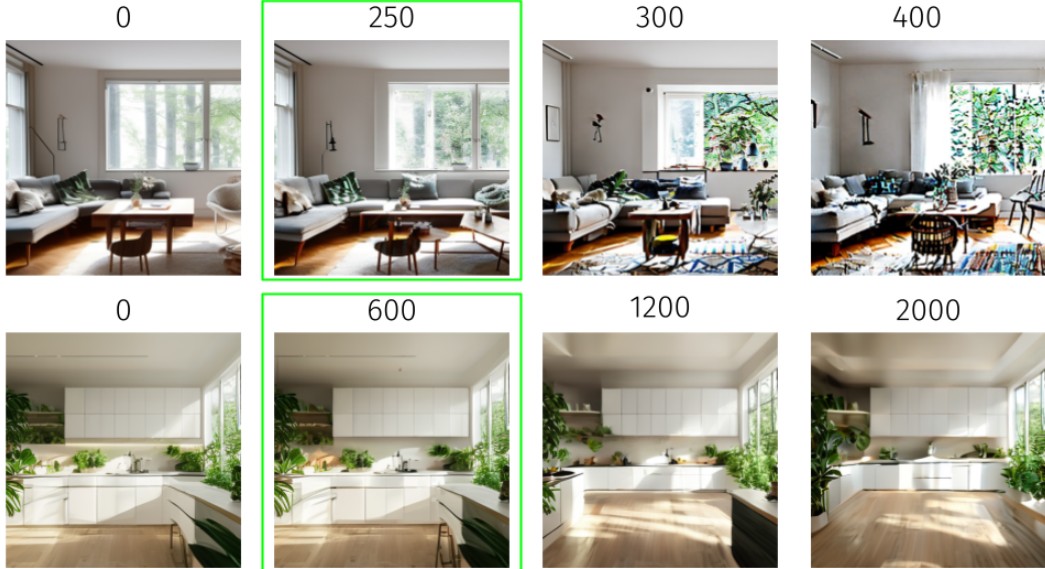

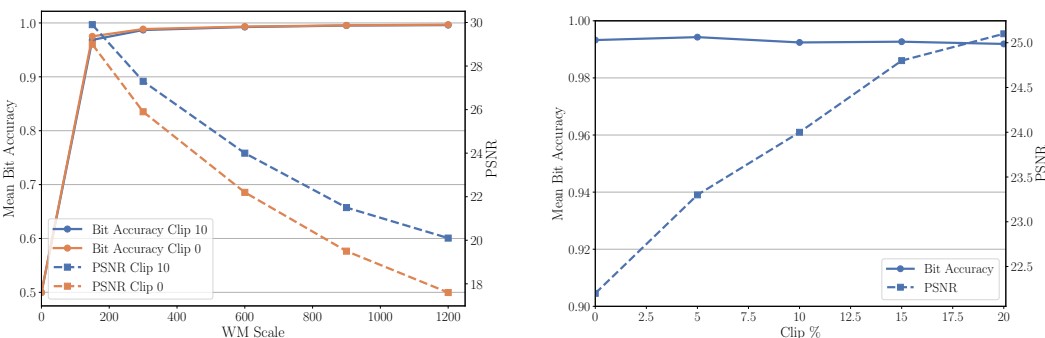

Figure 8: Example of images generated with our `G-SSig` and `G-VS` for different values of $\omega$. The other parameters are referenced in table 4. From Top to Bottom: with Stable Diffusion 2 with `G-SSig` and Sana with `G-VS`. The values of $\omega$ are intentionally exaggerated to highlight visible artifacts. Our choice appears framed in green.

Figure 9: Ablation study on the effect of WM Scale $\omega$ and Clip% $\tau$ for Sana `G-VS`images. On the left, the figure reports how Bit Accuracy and PSNR vary as a function of the WM Scale, both with and without clipping. On the right, Bit Accuracy and PSNR are shown as a function of $\tau$ for a fixed $\omega$.

## C  FALSE ALARM ANALYSIS

The most important feature of watermarking, as far as zero-bit detection is concerned, is that the probability of false alarm or, equivalently, the $p$-value are certified, contrary to forensics approaches where the false alarm rates are only empirically evaluated.

The watermarking literature considers two definitions of the probability of false alarm. Suppose that the watermark detector first extracts from an image $x$ a feature $\phi(x)$ and compares it to a secret vector $u$ by a score function $s(\phi(x), u)$. We assume here that the higher the score, the more likely the image is watermarked. This score is converted into a $p$-value that is the probability that an image not watermarked with this secret key (hypothesis $\mathcal{H}_0$) produces a score greater than $s(\phi(x), u)$. The image $x$ is deemed watermarked if the corresponding $p$-value is lower than the required probability of false alarm: $p < P_{FA}$

| WM scheme | ‖ | PSNR | LPIPS | CLIP |
|---|---|---|---|---|
| G-VS | ‖ | 23.5 | 0.07 | 0.348 |
| G-VS last 15 | ‖ | 28.3 | 0.02 | 0.348 |
| G-VS last 10 | ‖ | 30.6 | 0.01 | 0.347 |
| G-VS last 5 | ‖ | 35.8 | 0.004 | 0.348 |

Table 5: Quality metrics of images generated with Sana using G-VS for different numbers of guidance steps.

| LDM | WM | ‖ | CLIP ($\uparrow$) | PSNR ($\uparrow$) | LPIPS ($\downarrow$) | ‖ | Capacity($\uparrow$) | $-\log_{10}(\mathrm{P}_{\mathrm{FA}})$ @ $\mathrm{P}_{\mathrm{D}} = 0.9$ |
|---|---|---|---|---|---|---|---|---|
| Sana | G-VS | ‖ | 0.346 | 23.5 | 0.07 | ‖ | **207.5 (+28.8)** | **96.4 (+49.2)** |
| Sana | G-VS w/o simplification 1 | ‖ | 0.170 | 8.8 | 0.67 | ‖ | 140.7 | 49.1 |

Table 6: Comparison of image quality and detectability metrics for 200 Sana G-VS images, with and without Simplification 1.

A first way to compute this $p$-value is to model the distribution of $\phi(X)$ where $X$ denotes a random un-watermarked image:

$$p_X(x, u) = \mathbb{P}(s(\phi(X), u) > s(\phi(x), u) | X \text{ random image}). \tag{11}$$

That is, we assume that the detector uses a fixed key, and we control the probability of false alarm for this specific key. This approach is relevant in applications where the secret vector $u$ is unique, like in copy protection. Yet, it is challenging to accurately model the distribution of $\phi(X)$ due to the vast diversity of images.

A second way is to consider that the image is fixed, but that the secret vector is random. This is typically relevant for applications like traitor tracing, where a secret key is randomly drawn for each user.

$$p_U(x, u) = \mathbb{P}(s(\phi(x), U) > s(\phi(x), u) | U \text{ random secret vector}). \tag{12}$$

This approach is usually more accurate as the distribution of the secret vector is known exactly.

This appendix determines to which extent these approaches are suitable for the watermarking schemes considered in this paper.

## C.1   APPROACH 1: $X$ IS A RANDOM IMAGE

### C.1.1   STABLE SIGNATURE, TRUSTMARK, AND VIDEOSEAL

Our method resorts to a pre-trained watermark decoder extracting a feature $\phi(x) \in \mathbb{R}^M$ from an image $x$. This feature is then compared to a reference secret signal $u \in \mathbb{R}^M$ by a cosine similarity. The computation of the $p$-value equation 7 makes the assumption that $\phi(X)$ is a centered random vector with an isotropic distribution in space $\mathbb{R}^M$ when $X$ is a random non-watermarked image.

**Original implementation**   A first experiment investigates on (1) the unbiasedness and (2) the isotropy of the decoder's output by estimating its bias and covariance matrix. The output of each detector is computed over $n = 10^6$ images from the ELSA-D3 dataset. The images are resized with a bilinear filter to the size expected by the detector: $512 \times 512$ for Stable-Signature, $245 \times 245$ for Trustmark and $256 \times 256$ for VideoSeal.

$$b_\phi = \frac{1}{n} \sum_{i=1}^{n} \phi(x^{(i)}), \tag{13}$$

$$\mathbf{\Sigma}_\phi = \frac{1}{n-1} \sum_{i=1}^{n} (\phi(x^{(i)}) - b_\phi)(\phi(x^{(i)}) - b_\phi)^\top, \tag{14}$$

with $x^{(i)}$ being the $i$-th image. A second experiment bombards the decoder with real images (un-watermarked) and computes the cosine scores. Then, with a varying threshold ranging from -1 to 1, it compares the theoretical equation 7 and empirical probabilities of false alarm.

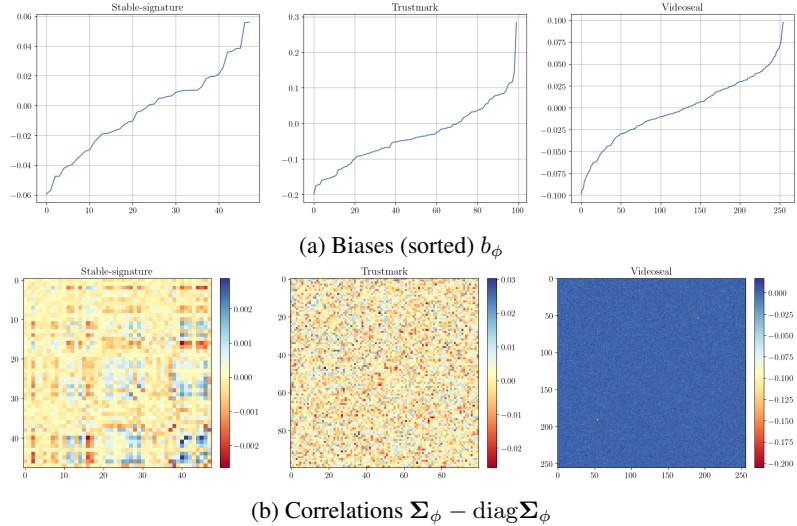

(a) Biases (sorted) $b_\phi$

(b) Correlations $\mathbf{\Sigma}_\phi - \mathrm{diag}\mathbf{\Sigma}_\phi$

Figure 10: Estimated biases and covariance matrix computed over $n = 10^6$ images of ELSA-D3 for each detector. The biases are sorted in ascending order.

Figure 10 shows that the official implementations of the three studied detectors – Stable-Signature (without correction), Trustmark and Videoseal – do not comply with the assumption: the outputs of their detector are either highly biased, highly correlated or both. *Such biases are already reported and corrected in the original Stable-Signature paper Fernandez et al. (2023), but no study of other detectors was performed to the best of our knowledge.*

The implication may lead to an unfair benchmark of the watermark. Figure 11 first plots the empirical *vs.* the theoretical probabilities of false alarm over $10^7$ scores (10 random reference signals $u$ and $10^6$ images). At first sight, there is a clear match. However, the situation might differ for a fixed reference signal. The worst case is to dishonestly set $u = \mathrm{sign}(b_\phi)/\sqrt{M}$, then the robustness of the watermark happens to be big but the probability of false alarm is not valid at all: the empirical probability is around $10^{-2}$ when these decoders claim a theoretical probability of false alarm of $10^{-6}$.

**Correction with whitening** Stable Signature suggests one way to correct this with a whitening process Fernandez et al. (2023). We apply this patch to TrustMark and VideoSeal. The experiments in our work are always performed with a whitened version of these official detectors. We now describe the whitening protocol. For each detector, the feature vector is computed over $n = 10^6$ images from the ELSA-D3 dataset. We then compute the empirical bias equation 13 and covariance matrix equation 14. Finally, we compute the Cholesky decomposition $\mathbf{\Sigma}_\phi = \mathbf{L}_\phi \mathbf{L}_\phi^\top$. The whitened detector is then defined as:

$$\phi_w(x) = \mathbf{L}_\phi^{-1}(\phi(x^{(i)}) - b_\phi). \tag{15}$$

Our validation computes the empirical and theoretical probabilities of false positive on two datasets of one million images: ELSA-D3 and MIRFLICKR. Figure 11 reports the results. The empirical probability of false alarm now matches the theoretical one, whatever the reference signal $u$ used. The limitation of this study is that the agreement with the theoretical model is only verified up to a probability of false alarm in the order of $10^{-6}$. A lower level would require a number of images that is out of reach. Note, however, that the agreement is very precise, well below the estimation uncertainty $\gamma = \pm\sqrt{-\frac{1}{2n}\log(\alpha/2)}$ with probability $1 - \alpha$.

### C.1.2 TREE-RING, RoBIN AND GAUSSIAN-SHADING

In Tree-Ring (Wen et al., 2023), RoBIN (Huang et al., 2025) and Gaussian-Shading (Yang et al., 2024b), the watermark detector is not learned but 'hand-crafted'. From an image $x$, the reverse diffusion process estimates a seed $\hat{z}_T$, its Fourier transform $F(\hat{z}_T)$ is compared to the secret signal $u$ over a given mask region $\mathcal{M}$ in the Fourier domain symmetric around $(0,0)$. The final score is

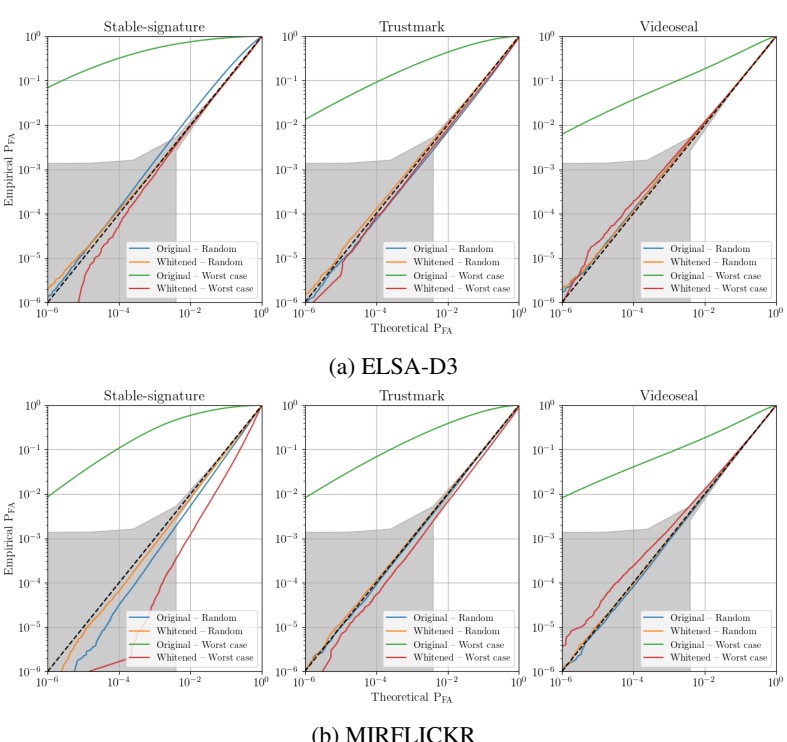

(a) ELSA-D3

(b) MIRFLICKR

Figure 11: Empirical probability of false alarm of whitened and non-whitened detectors computed over $n = 10^6$ images on two datasets, as a function of the theoretical probability of false alarm equation **??**. Random: measured over 10 random reference vector $u$, Worst case: reference vector $u$ set according to the bias $b_\phi$. Sound detectors should output values matching the dotted black lines.

computed as an Euclidean distance: $s = \hat{\sigma}^{-2} \sum_{i \in \mathcal{M}} |F(\hat{z}_T)_i - u_i|^2$, where $\hat{\sigma}^2$ is the estimated power of $F(\hat{z}_T)$ (see details in Wen et al. (2023)).

Now, to turn a score $s$ into a $p$-value, the authors of Tree-Rings assume that from a random non-watermarked image $X$, the reconstructed seed $\hat{Z}_T$ and hence its Fourier transform follow the i.i.d. Gaussian distribution (real or complex, respectively) of variance $\hat{\sigma}^2$. Therefore, the score is distributed as a noncentral $\chi^2_{k,\lambda}$ with degree of freedom $k = |\mathcal{M}|$ (the number of components selected by the mask region $\mathcal{M}$) and non-centrality $\lambda = \hat{\sigma}^{-2} \sum_{i \in \mathcal{M}} |u_i|^2$. This rationale allows to compute the $p$-value.

However, our simple experiments show that the $p$-value $p_X$ is not valid. Here is the outcomes of our investigation:

- Even when $\hat{Z}_T$ is drawn according to the i.i.d. Gaussian distribution, the $p$-value is not rigorous since their pdf is not flat (see Fig. 12 - Top). First, the degree of freedom is not $k = |\mathcal{M}|$ due to the Hermitian symmetry in the Fourier domain. Indeed, one should compute the Euclidean distance over a half of the mask region. Also, in the original implementation, the reference signal does not comply with the Hermitian symmetry. Once patched, the $p$-value is uniformly distributed as it should be (see Fig. 12 - Bottom).

- When the reverse diffusion estimates a seed from a real image, the empirical $p$-value is not uniformly distributed, be it computed with (Fig. 12 - Bottom) or without (Fig. 12 - Top) our patch. This shows that the assumption (estimated seed i.i.d. Gaussian distributed), on which the computation of the $p$-value is based, does not hold. Moreover, the computed $p$-values are abnormally low.

The same comment applies to (RoBIN Huang et al. (2025)) with even more divergence because the reconstructed latent $z_t$ at step $t$ is even less Gaussian distributed with components heavily correlated, which completely spoils the computation of the $p$-value.

On the other hand, Gaussian-Shading does not compute the score directly on the estimated seed. Using a secret key and a stream cipher, it converts the estimated seed into a binary sequence $\hat{u}$. Following the statistical guarantees offered by the stream cipher, it can be assumed safely that each bit in the binary sequence is independent and uniformly distributed. This allows the computation of a sound p-value using the score:

$$s(\hat{U}, u) = \sum_{i=1}^{M} \left[ \hat{U}_i = u_i \right]. \tag{16}$$

Under $\mathcal{H}_0$, each element of the sum is i.i.d. Bernoulli distributed $\mathcal{B}(0.5)$, hence a the $p$-value is given by:

$$p_X = I_{\frac{1}{2}} \left( s(\hat{U}, u), M - s(\hat{U}, u) + 1 \right), \tag{17}$$

where $I_x(a, b)$ is the regularized incomplete beta function.

This discussion reveals the difficulties bound to this first approach: the soundness of $p$-values $p_X$ computed for post-hoc schemes can only be guaranteed up to a level – here $10^{-6}$ – which depends on the amount of un-watermarked images we can test. Furthermore, we cannot offer sound $p$-values $p_X$ for *in-gen* watermarking techniques, except for Gaussian-Shading.

### C.2 APPROACH 2: $U$ IS A RANDOM VECTOR

The second approach keeps the image fixed and the secret vector random. In other words, the computation of the $p$-value $p_U$ only holds for the image $x$ under scrutiny. The advantage is that the secret vector $U$ distribution is known and easy to sample from. In the worst case, it is thus possible to estimate the $p$-value equation 12 through Monte-Carlo methods. In the best case, a closed-form formula exists, and this is indeed the case for the considered watermarking techniques.

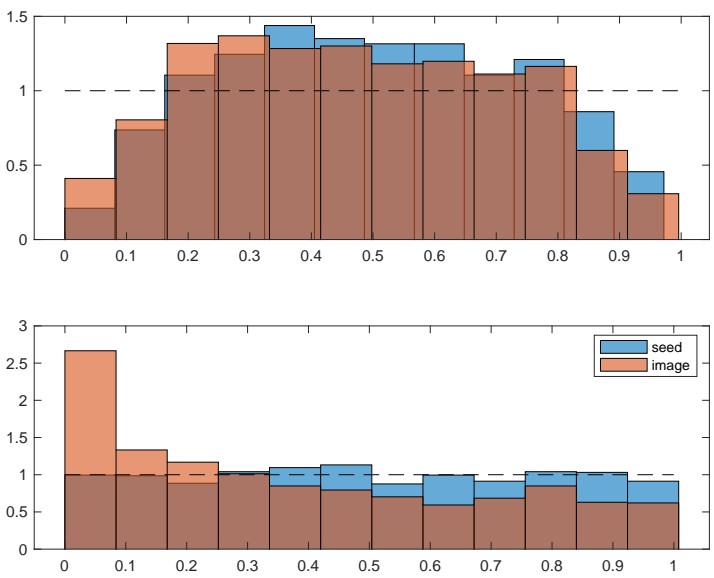

Figure 12: Empirical pdf of the $p$-values of Tree-Ring. Top - with the original code. Bottom - with our patch. Blue - with a seed randomly drawn from a Gaussian distribution, Orange - with a seed reconstructed from an image (1,300 images from MIRFlickR).

### C.2.1 STABLE SIGNATURE, TRUSTMARK, VIDEOSEAL, AND GAUSSIAN-SHADING

In the case of post-hoc schemes and Gaussian-Shading, the score is symmetrical between the features extracted from the image $\phi(x)$ and the secret vector $u$. Moreover, the secret vector and the feature share the same statistical model. The reasoning is thus completely identical, and we find back the same $p$-values formulas (7) and (17). In other words, $p_X(x, u) = p_U(x, u)$.

### C.2.2 TREE-RING

We only treat the case where Tree-Ring uses the *ring* method for the watermark signal – see Section 3.3 in Wen et al. (2023). Let the secret vector $U$ be composed of $J$ rings. Each ring duplicates the random variable $U_j$ over the $r_j$ components on a first half disk, and duplicates its conjugate transpose $\bar{u}_j$ on the second half disk to comply with the Hermitian symmetry. The random variable $U_j$ is sampled from a complex standard Gaussian $r_j \sim \mathbb{CN}(0; I_{r_j})$. Let $\hat{z}$ be the seed estimated by Tree-Ring from an image $x$. The detection score is computed as:

$$s(\hat{z}, U) = \sum_{j=1}^{J} \sum_{i=1}^{D_j} |\hat{z}_{i,j} - U_j|^2 = \sum_{j=1}^{J} D_j |U_j - \lambda_j|^2 - c \quad (18)$$

$$\text{with } \lambda_j = \frac{\sum_{i=1}^{D_j} \hat{z}_{i,j}}{D_j} \in \mathbb{C}, \text{ and } \quad c = \sum_{j=1}^{J} \left( \frac{|\lambda_j|^2}{D_j} - \sum_{i=1}^{D_j} \hat{z}_{i,j}^2 \right). \quad (19)$$

This shows that, for a fixed seed $\hat{z}$, the distribution of $s(\hat{z}, U)$ is a weighted sum of non-central chi-square random variables plus the offset $c$. The probability of false-alarm is obtained by computing the left, finite tail, of this distribution. This can be done up to any degree of accuracy using Ruben's method (Ruben, 1962, Eq.(5.26)).

## D MORE RESULTS

### D.1 BIT-ACCURACY AND $p$-VALUES

Figures 13 and 14 present a comparison between our guidance-based watermarking method and the original approach across all combinations of three models (Stable-Diffusion 2, Flux, Sana) and two watermarking techniques (Stable-Signature, VideoSeal). Stable-Signature is an in-generation watermarking method working by fine-tuning the VAE at the end of the diffusion. VideoSeal is post-hoc method. Our guidance approach improves the detectability using Stable-Signature detector without fine-tuning the VAE. It transforms VideoSeal into in-generation method, achieving high performance and robustness. All results were computed over 1,000 images per method, using the guidance parameters reported in Table 4. We perform the guidance using augmentations known by the detector. The transform set was: Contrast $\times 2.0$, Brightness $+0.2$, Crop 50%, JPEG 50, JPEG 80. We use the differentiable pseudo JPEG transform from the library Kornia.

To ensure a fair comparison across all combinations, we evaluate the robustness of each method against the transformation sets defined for Stable-Signature and VideoSeal. This time, we use the real JPEG lossy compression as implemented in library Augly. Our guidance approach consistently outperforms the original method when using either the Stable-Signature or VideoSeal detector.

### D.2 COCO DATASET

In the main paper, experiments were performed for a realistic set of prompts, usually leading to pretty detailed images, which are quite amenable to watermarking. To stress-test our approach, we repeated the experiment on 200 captions from the standard COCO dataset(Lin et al., 2014). This dataset contains far simpler and less diverse prompts. It leads to images with, on average, a lot less content and details which is a good test of the limit of the watermarking approach. We report these results in Figure 15. Once again, we reliably outperform the baseline post-hoc schemes, though by a smaller margin for Sana. If we study images which lead to the worst detectability on our guidance method, we find images that are characteristically difficult to watermark: little content, almost no texture and flat-colored skies – see Figure 16.

### D.3 DEPENDENCY ON THE CONTENT

Figure 17 shows the mean difference between watermarked and non-watermarked images for `G-VS`, Tree-Rings, and Gaussian-Shading, following the procedure suggested in Yang et al. (2024a).Note that we scaled ten times the signal from `G-VS` for better visualization. This averaged signal is typically used to perform average attacks or copy attacks against seed-based watermarking schemes. Distinct patterns emerge when averaging the extracted signals of Gaussian-Shading and Tree-Rings for a fixed key, whereas the signal extracted from the guidance method quickly converges to zero everywhere resembles. This observation empirically indicates that the guidance watermarking is content-dependent, in contrast to seed-based methods.

## E METHODOLOGY COMPARISON WITH OTHER WATERMARKING SCHEMES

Figure 21 illustrates the methodological differences between watermarking approaches at both embedding and detection stages. Post-hoc methods embed the watermark after image generation using a dedicated embedder. Since guidance transforms a post-hoc watermarking scheme into an in-generation one, **both ultimately rely on the same detection procedure**. The output image is fed to a detector, which returns a vector $\phi(x_o)$. This vector is then compared to the secret vector $u_m$ to compute various detection metrics.

In contrast, seed-based methods watermark the seed itself and use it with the diffusion model to generate the final image. At detection time, a reverse diffusion process is applied to the image to recover an approximation of the watermarked seed, denoted $z'_T$. As reported in Table 3, this reverse process requires performing several diffusion steps. Once again, the estimated seed is compared to a secret vector $u_m$ – e.g. the ring patterns for Tree-Rings - to yield the detectability metrics.

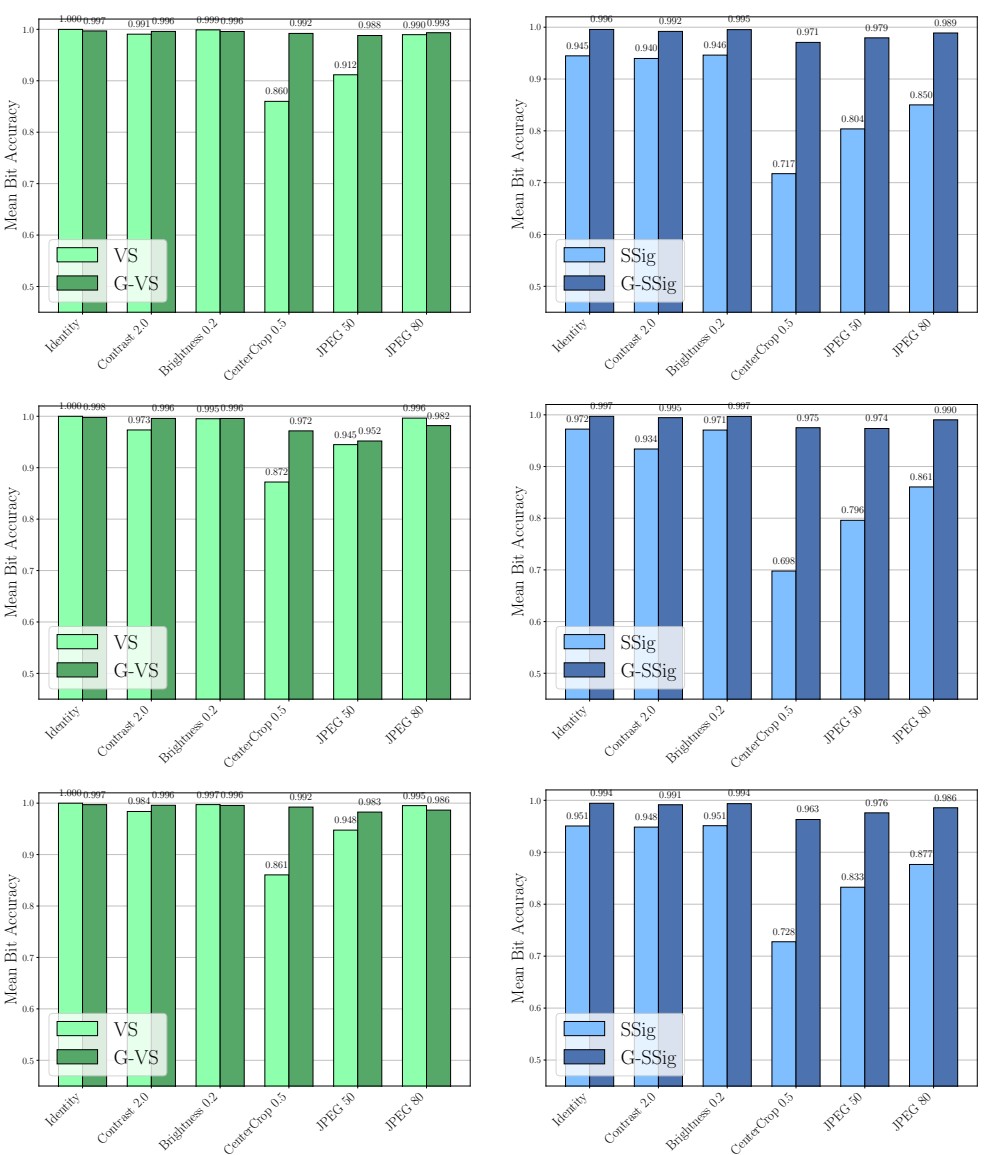

Figure 13: Mean Bit Accuracy for all combinations of diffusion model and watermark decoder. Higher is better.

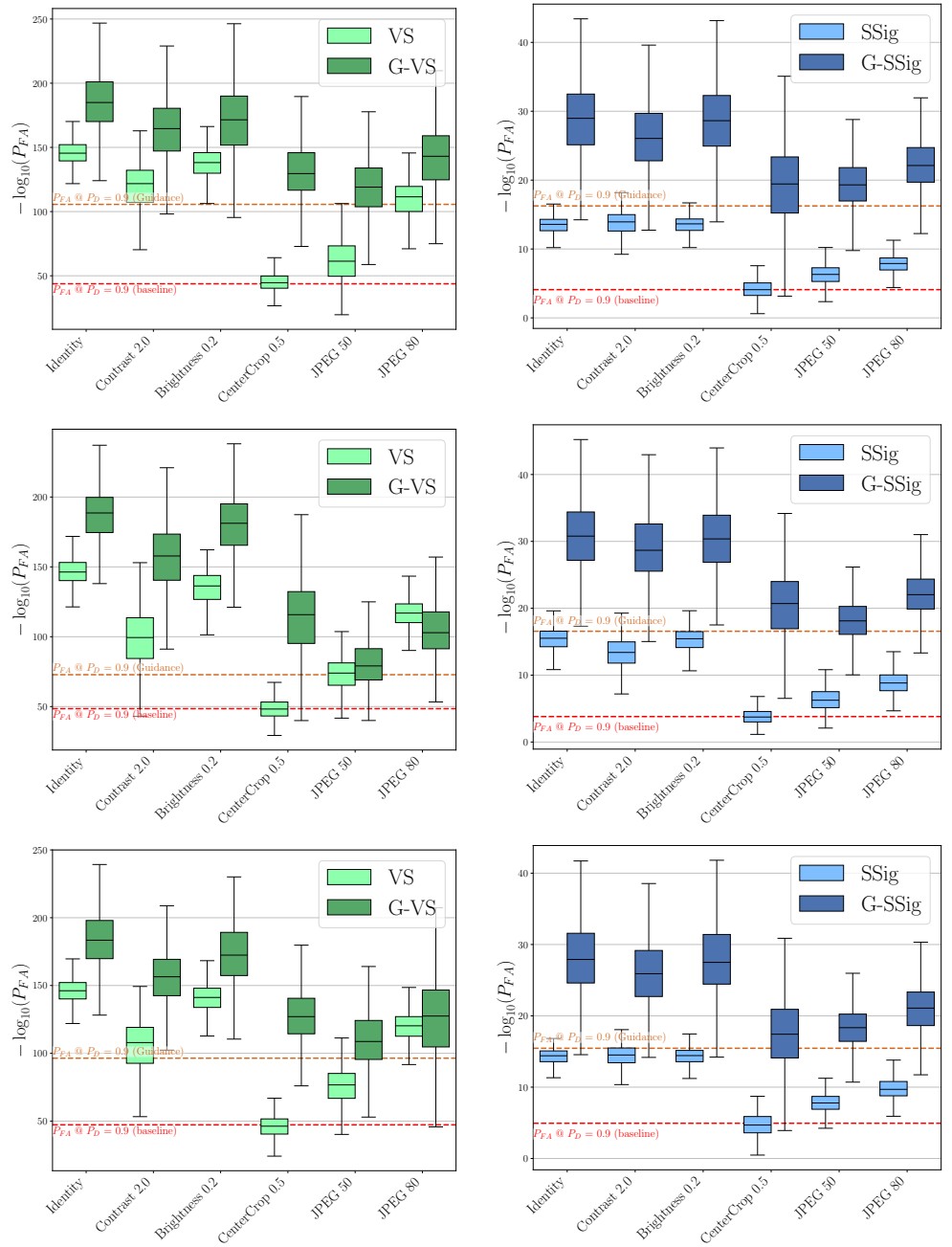

Figure 14: $-\log_{10}(P_{FA})$ for all combinations of diffusion model and watermark detector. Higher is better. Thresholds are $-\log_{10}(P_{FA})$ @ $P_D = 0.9$ for the baseline method (red) and guidance (brown).

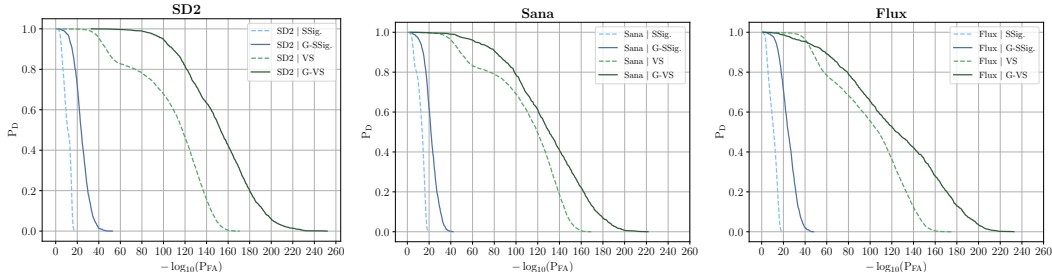

Figure 15: Repeated experiment for 200 images generated form the COCO captions of Figure 5.

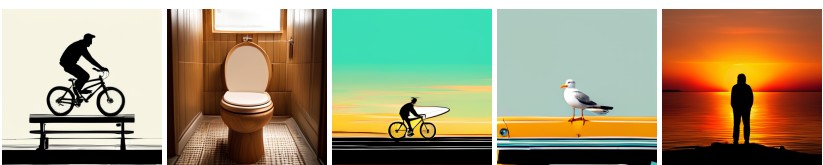

Figure 16: Examples of MSCOCO images generated by Sana with our `G-SSig` giving birth to high $p$-values.

Guidance and post-hoc detections are instantaneous, requiring only a single call to the detector, whereas seed-based methods require a higher detection cost due to the additional diffusion steps needed. The trade-off is that our guidance methods requires more diffusion step during the embedding phase than seed-based methods.

## F    COMPUTATIONAL RESOURCES

All experiments were conducted using NVIDIA A100 and L40S GPUs with 40 GB and 45 GB of memory respectively. We generate between 200 and 300 images per hour, depending on the choice of GPU, model, detector, and hyperparameters used for guidance. Significant efforts were made to optimize memory usage and batch sizes in order to fully utilize available GPU resources and reduce energy costs. The code is available on the repository `XYZ` (provided upon acceptance).

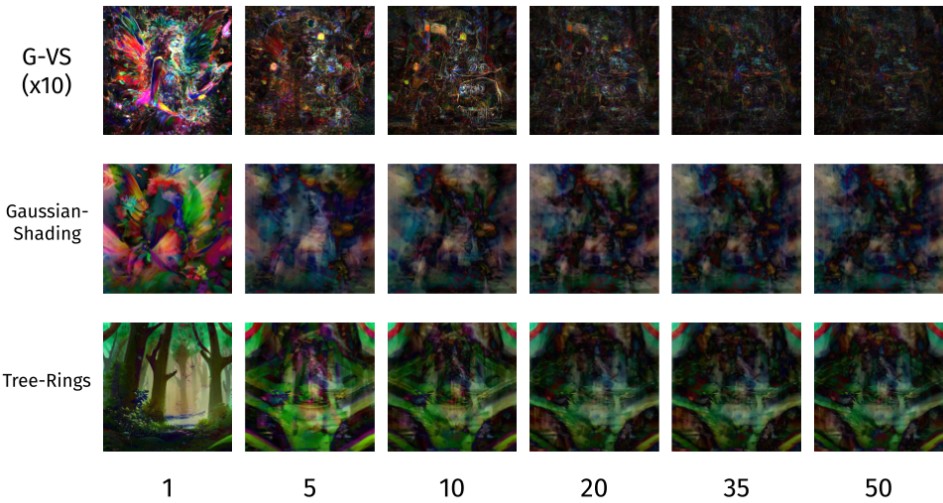

Figure 17: Averages of the residues for Sana using three in-generation watermarking schemes, `G-VS`, Gaussian-Shading, and Tree-Rings, for different numbers of extracted residues. For `G-VS`, the averaged signal is multiplied by 10 to make it visually discernible.

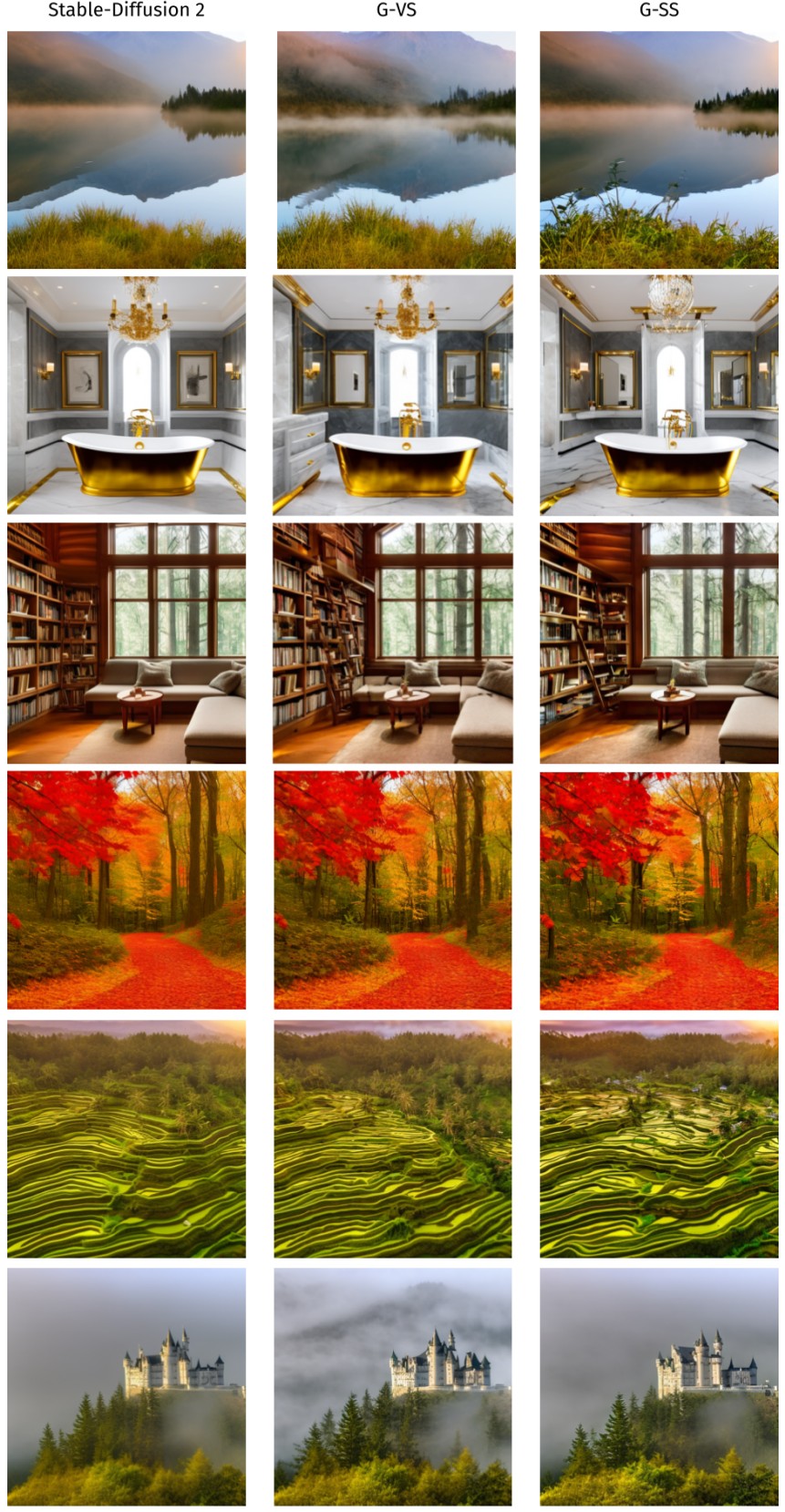

Figure 18: Images generated by Stable-Diffusion without and with our watermark embedding for `G-VS`, and `G-SSig`. The resulting images remain semantically very similar to each other.

| FLUX | G-VS | G-SS |
|------|------|------|

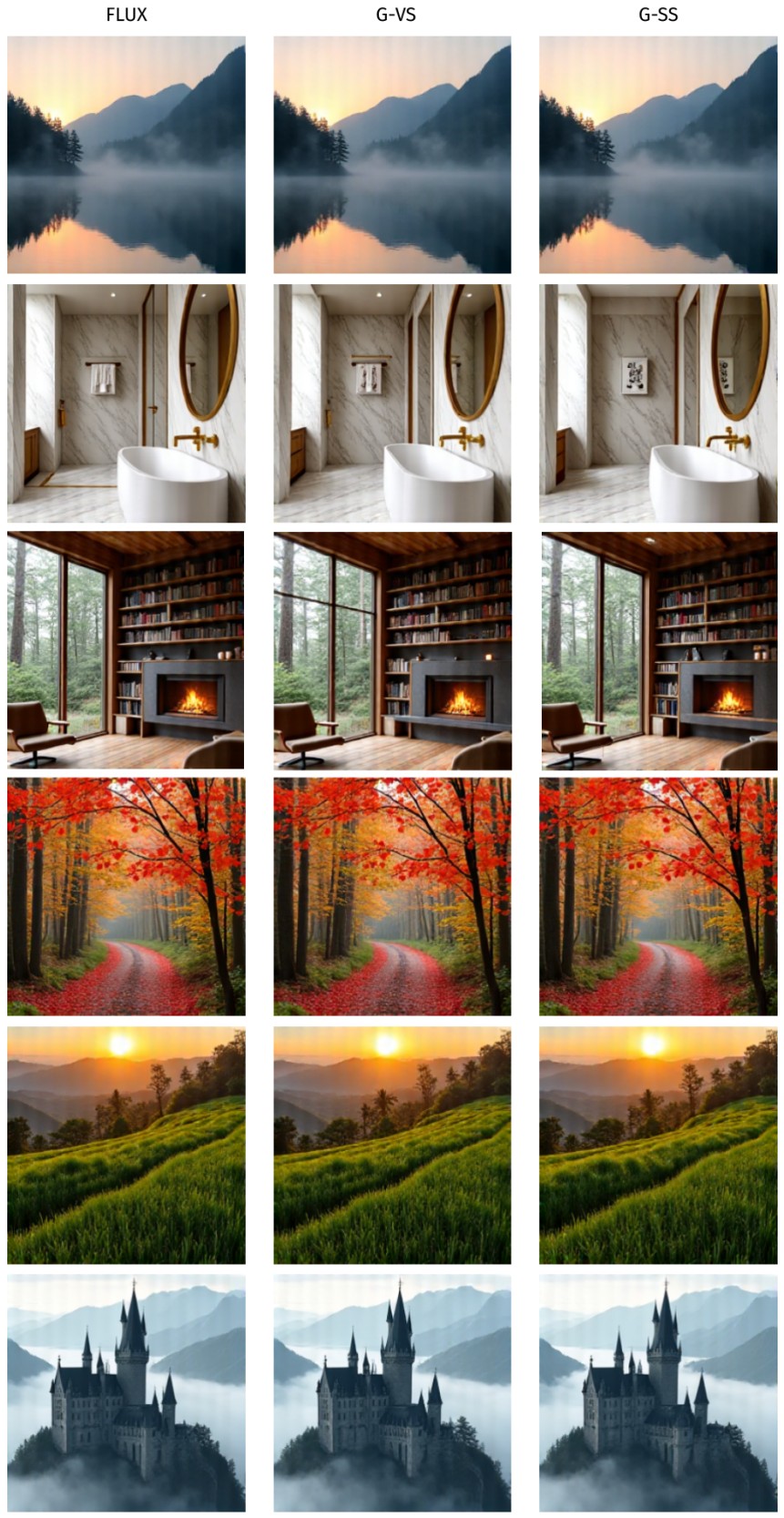

Figure 19: Images generated by Flux without and with our watermark embedding for `G-VS`, and `G-SSig`. The resulting images remain semantically very similar to each other.

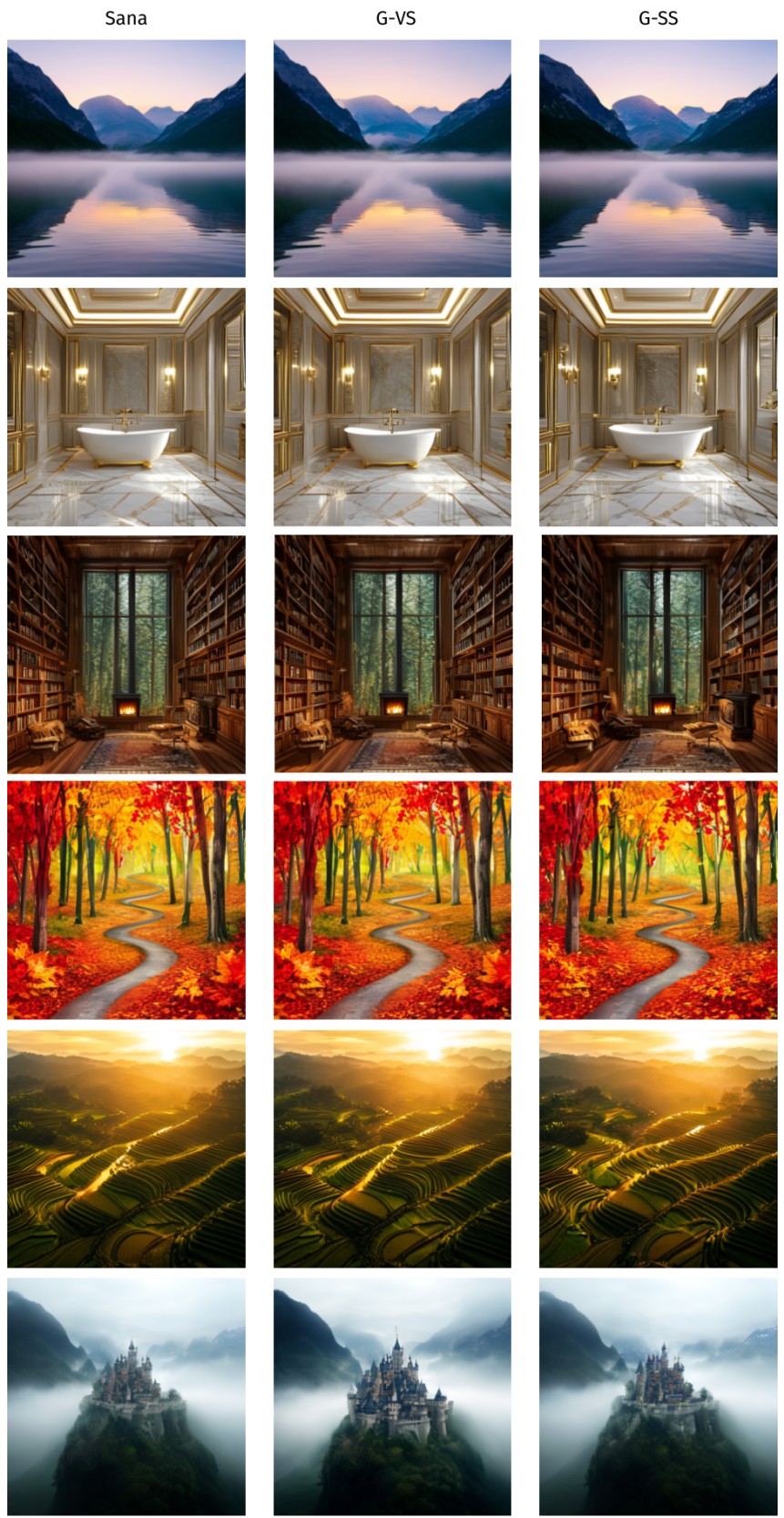

Figure 20: Images generated by Sana without and with our watermark embedding for `G-VS`, and `G-SSig`. The resulting images remain semantically very similar to each other.

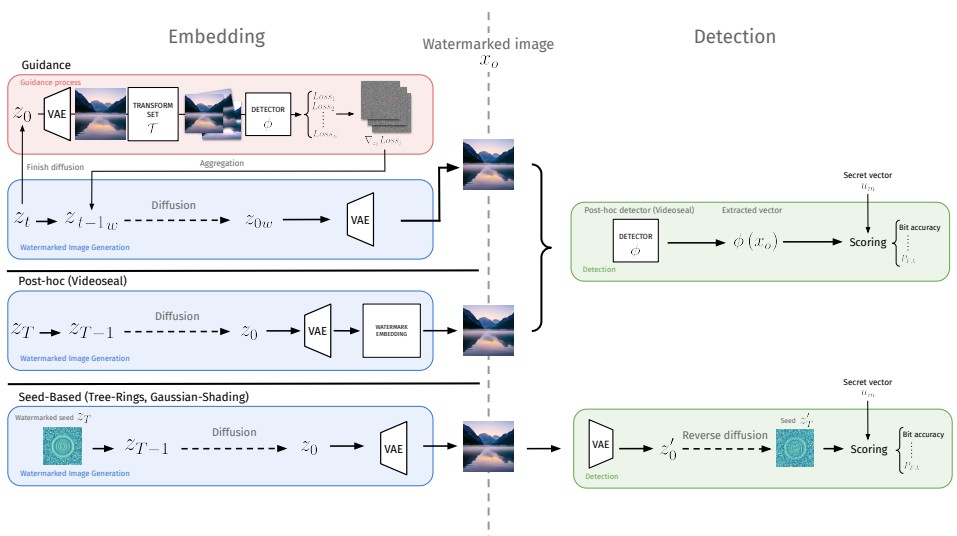

Figure 21: System-level diagram of the comparison of different watermarking methods.

# G LICENCES

We ensure that datasets, pretrained diffusion models and watermarking detectors used in this work are credited and used under the terms of their respective licenses.

## DATASETS

We use the 2014 COCO dataset Lin et al. (2014) for the prompts and the images to compute of FID. We use the Stable-Diffusion-Prompts dataset from Hugging Face for realistic prompts. We use ELSA-D3 and MIRFLICKR for the whitening.

### COCO

- Source: `https://cocodataset.org/#home`
- License: CC BY 4.0

### Stable-Diffusion-Prompts

- Source: `https://huggingface.co/datasets/Gustavosta/Stable-Diffusion-Prompts`
- License: Unknown

### ELSA-D3

- Source: `https://huggingface.co/datasets/elsaEU/ELSA_D3`
- License: CC BY 4.0

### MIRFLICKR

- Source: `https://press.liacs.nl/mirflickr/mirdownload.html`
- License: CC BY 4.0

## PRETRAINED DIFFUSION MODELS

We use the pre-trained diffusion models to generate images Rombach et al. (2022b); Black Forest Labs (2024); Xie et al. (2024), watermarked images with our guidance and watermarked images with other methods. We used their Hugging Face implementation. All models used in this work were modified in accordance with the terms of their respective licenses to incorporate our guidance method.

### Stable-Diffusion 2.1-base

- Source: `https://huggingface.co/stabilityai/stable-diffusion-2-1-base`
- License: CreativeML Open RAIL++-M License (v24 novembre 2022)

### FLUX 1.0 dev

- Source: `https://huggingface.co/black-forest-labs/FLUX.1-dev`
- License: FLUX.1 [dev] Non-Commercial License

### Sana

- Source: `https://huggingface.co/Efficient-Large-Model/Sana_600M_512px`
- License: NVIDIA License

WATERMARKING DETECTORS

We use the watermark detectors Fernandez et al. (2023); Bui et al. (2023); Fernandez et al. (2024a) to generate watermarked images with our guidance and with their original embedding methods.

### Stable-Signature

- Source: `https://github.com/facebookresearch/stable_signature`
- License: CC BY-NC 4.0

### TrustMark

- Source: `https://github.com/adobe/trustmark`
- License: MIT License

### VideoSeal

- Source: `https://github.com/facebookresearch/videoseal`
- License: MIT License

ATTACKS

We use the Python library Kornia for image attacks, as it provides differentiable versions of many transformations, a requirement for the guidance method. We rely on the AugLy library to apply real JPEG compression, as opposed to Kornia's differentiable approximation, in order to perform exact evaluations.

### Kornia

- Source: `https://github.com/kornia/kornia`
- License: Apache-2.0 License

### Augly

- Source: `https://github.com/facebookresearch/AugLy`
- License: MIT License

