# OpenReview forum: "Guidance Watermarking for Diffusion Models"
_ICLR.cc/2026/Conference — ICLR 2026 Poster_

### Official Review · Reviewer_jcck · 2025-10-21

**Soundness:** 3
**Presentation:** 2
**Contribution:** 2
**Rating:** 4
**Confidence:** 3

**Summary:**

This work proposes a guidance module that could be applied during the diffusion process to embed a watermark. It relies on a pretrained watermark decoder/detector (such as Stable Signature) to compute the loss that is needed for the guidance. The main idea is to add a gradient term to the noise during each diffusion step and the gradient term is computed with respect to the watermark. It gives experimental results to show the effectiveness of this method.

**Strengths:**

+ The guidance module could strengthen existing pretrained baseline watermarking method such as Stable Signature and VideoSeal, as shown in their experiments in Table 1. More importantly, it extends baselines to be more robust agains unknown augmentations.

+ Experimental results on various diffusion models are provided: SD2, Flux, and Sana.

**Weaknesses:**

- With all those formulas floating around, it is challenging to grasp the whole picture of this work and the additional contribution they bring to existing watermarking pipeline. (See my suggestion below)

- Choice of the guidance strength hyperparameter adds complexity to the watermarking application.

- It is not completely fair to claim that your work "does not necessitate any retraining of the diffusion model" and compare against TreeRing and GaussianShading (both trainning-free), when your method actually relies on a well-trained watermark detector (e.g., Stable Signature).

- The robustness results are not consistently better than GaussianShading in Table 2.

**Questions:**

- I am confused about the Motivations section. If your motivation is that the spectral signature could be exploited to remove the watermark, why not address this issue in your evaluation?

- Explain more on why the right pattern is better than the left in Figure 1.

- A system view (probably a system/pipeline figure) is necessary to understand their method and the additional module/steps they bring into the watermark embedding/detection pipeline, e.g., compared to the baseline Stable Signature.

- It is not very clear how detection/decoding is performed. Maybe (7) gives some info on detection, but what about decoding? How do we get the watermark message from an image?

- In Table 1, is any image attack used in the evaluation for robustness? If so, what are they? If not, you should add robustness comparison with SSign and VS in the Table 2.

- The basic image distortions in Table 2 seem not sufficient, more to add: e.g., gaussian blur, rotation.

- Confused about your claim: "ours does not require extra steps for decoding making it up to 50 times faster at detection
time." I think this is also because the decoding procedure is not clearly explained in the paper. For GuassinShading/TreeRings, we need to convert an image to latent noise for detection. Are u claiming that your method does not require such conversion? If so, how do you directly extract watermark from an image?

---

> ### Author Response · Authors · 2025-11-20
> **Part 1/2**
>
> We are grateful to the reviewer for their thoughtful evaluation. We value their acknowledgment of the robustness of our method.
>
> ## Weaknesses
>
> > With all those formulas floating around, it is challenging to grasp the whole picture of this work and the additional contribution they bring to existing watermarking pipeline. (See my suggestion below)
>
> Thank you for the remark, we included a system-level diagram in the paper (new Figure 1) in the paper to improve clarity, as suggested in the questions.
>
> > Choice of the guidance strength hyperparameter adds complexity to the watermarking application.
>
> Thank you for the remark. We included a more complete ablation studies in Appendix B.1 to illustrate their impact.
>
> | **Clip %** | **WM Scale** | PSNR | LPIPS | CLIP | Bit Accuracy |
> |---|---|---|---|---|---|
> | 0% | 0 | Inf | 0.0 | 0.344 | 0.500 |
> |  | 150 | 29.0 | 0.03 | 0.344 | 0.975 |
> |  | 300 | 25.9 | 0.05 | 0.345 | 0.989 |
> |  | 600 | 22.2 | 0.10 | 0.344 | 0.993 |
> |  | 900 | 19.5 | 0.18 | 0.342 | 0.996 |
> |  | 1200 | 17.6 | 0.24 | 0.340 | 0.997 |
> |---|---|---|---|---|
> | 10% | 0 | Inf | 0.0 | 0.344 | 0.500 |
> |  | 150 | 29.9 | 0.02 | 0.344 | 0.969 |
> |  | 300 | 27.5 | 0.04 | 0.344 | 0.987 |
> |  | 600 | 24.0 | 0.07 | 0.345 | 0.992 |
> |  | 900 | 21.5 | 0.12 | 0.345 | 0.995 |
> |  | 1200 | 20.1 | 0.16 | 0.342 | 0.996 |
> |---|---|---|---|---|
> | 0% | 600 | 22.2 | 0.10 | 0.344 | 0.993 |
> | 5% |  | 23.3 | 0.08 | 0.344 | 0.994 |
> | 10% |  | 24.0 | 0.07 | 0.345 | 0.992 |
> | 15% |  | 24.8 | 0.06 | 0.345 | 0.993 |
> | 20% |  | 25.1 | 0.06 | 0.345 | 0.992 |
>
> Increasing the watermarking guidance scale improves bit accuracy but also amplifies the deviation from the non-watermarked image, as reflected by a lower PSNR. The parameter clip % clips extreme values to prevent visual artifacts, increasing clip % results in a higher PSNR with only a minor reduction in bit accuracy.
>
> > It is not completely fair to claim that your work "does not necessitate any retraining of the diffusion model" and compare against TreeRing and GaussianShading (both trainning-free), when your method actually relies on a well-trained watermark detector (e.g., Stable Signature).
>
> Thank you for your remark. Our method is indeed retraining-free, as it only requires an off-the-shelf differentiable detector and a diffusion model. We focus on post-hoc methods because our goal is to convert them into in-generation methods, and the most effective post-hoc detectors are currently neural networks.
>
> > The robustness results are not consistently better than GaussianShading in Table 2.
>
> Thank you for the remark. To clarify, we report results for both G-VS and G-SSig. G-SSig performs guidance using the Stable-Signature decoder, which returns a 48-bit message; thus, a perfectly detected message would have a capacity of 48. We included this to demonstrate robustness to cropping, where Gaussian-Shading fails. For a fairer comparison, G-VS is more relevant, as it uses guidance with the Videoseal detector, which returns 256-bit messages, matching Gaussian-Shading. In this case, our guidance consistently outperforms Gaussian-Shading across all observed transformations and remains robust to cropping, unlike Gaussian-Shading. We add experiments to show the robustness of our guidance compare to in-gen methods for new augmentations (gaussian blur, rotation) and adversarial attacks (VAE purification, regeneration and averaging attacks) in Section 5.3 *Adversarial attacks*. In all thoses case, our guidance outperform existing in-gen seed-based attacks.
>
> ## Questions
>
> > I am confused about the Motivations section. If your motivation is that the spectral signature could be exploited to remove the watermark, why not address this issue in your evaluation?
>
> Thank you for your remark, we revised the writing of the motivation to clarify. The first observation was that Stable-Signature embeds the watermarking signal in a specific, high-frequency, band in the image. This fact was extensively studied in the original Stable-Signature paper, showing weakness to attacks specifically targeting this band. Our main motivation was thus to find a watermarking methods that would distribute the signal over a larger part of the spectrum, in order to improve its robustness to augmentations applying a low-pass filters (e.g. JPEG, downsampling) as well as VAE attacks.
> The question of security, as the reviewer rightfully points outs, is not the main focus of the paper. As such, we removed this point from the motivation section in order to focus on the main idea of designing a in-gen watermarking that better exploits the content of the generated images.

---

> > ### Author Response · Authors · 2025-11-20
> > **Part 2/2**
> >
> > > Explain more on why the right pattern is better than the left in Figure 1.
> >
> > The pattern on the right contains a watermarking signal which is well distributed over all of the spectrum. Most likely this should translate to a more robust scheme. Yet, this was only a motivation, and the rest of the paper tries to justify this idea by showing the high robustness and flexibility of our approach.
> >
> > > A system view (probably a system/pipeline figure) is necessary to understand their method and the additional module/steps they bring into the watermark embedding/detection pipeline, e.g., compared to the baseline Stable Signature.
> >
> > Thank you for the suggestion. We included a system-level diagram to make the presentation of the method clearer.
> >
> > > It is not very clear how detection/decoding is performed. Maybe (7) gives some info on detection, but what about decoding? How do we get the watermark message from an image?
> >
> > Thank you for the remark. Detection is performed by simply applying the off-the-shelf detector, as in the original papers. For example, we use the VideoSeal detector on our images, which returns a 256-dimensionnal embedding. This representation is converted to a 256-bit binary message using the sign of each vector element, as in the original method. This allows us to compute capacity, bit accuracy, and other metrics. This is a standard use case for a detector since the landmark HiDDeN paper [1]. Note that post-hoc decoders are typically neural networks, which our framework leverages directly without any retraining.
> >
> > [1] *Jiren Zhu, Russell Kaplan, Justin Johnson, and Li Fei-Fei. Hidden: Hiding data with deep networks. In ECCV, 2018.*
> >
> > > In Table 1, is any image attack used in the evaluation for robustness? If so, what are they? If not, you should add robustness comparison with SSign and VS in the Table 2.
> >
> > Thank you for the remark, we modified the caption of the Table 1 to make it clearer. As mentionned in paragraph 5.4, in Table 1, the reported values correspond to the mean over a batch of attacks. These augmentations are ones against which the original decoder is already robust. We show that applying guidance further improves robustness to these attacks. This might have been unclear due to the lack of information in the caption and has been dully addressed in the paper. The detailed results are also reported in the Appendix D, Figures 13 and 14.
> >
> > > The basic image distortions in Table 2 seem not sufficient, more to add: e.g., gaussian blur, rotation.
> >
> > Thank you for the suggestion. We extended this table with additional transformations for comparison. Our main point was to show that guidance remains robust to cropping, unlike other in-generation methods. Please find the results in Table 2. We summarize here the bit accuracy (Gaussian-Shading) or $-\log_{10}P_{FA}@P_D=90\%$ (Tree-Rings).
> >
> > | **WM scheme** | Gaussian Blur 3 | Rotation 90° |
> > |---|---|---|
> > | Gaussian-Shading | 216 | 0 |
> > | G-SSig | 25 | 27 |
> > | G-VS | **220** | **194** |
> > |---|---|---|
> > | Tree-Rings | 9.1 | 0.8 |
> > | G-SSig | 16.6 | 17.4 |
> > | G-VS | **150.3** | **100.7** |
> >
> > G-VS, guided through videoseal 256 bits detector is more robust than Gaussian-Shading and Tree-Rings for all the classic augmentations studied even for geometric transformations for which seed-based methods are not robust.
> >
> > > Confused about your claim: "ours does not require extra steps for decoding making it up to 50 times faster at detection time." I think this is also because the decoding procedure is not clearly explained in the paper. For GuassinShading/TreeRings, we need to convert an image to latent noise for detection. Are u claiming that your method does not require such conversion? If so, how do you directly extract watermark from an image?
> >
> > Thank you for your remark. To be clear, the detectors we use are VideoSeal and Stable-Signature. They are both neural networks. They take an image as input and directly return a binary message. This is in contrast to seed-based schemes such as Tree-Rings and Gaussian-Shading. The detectors of the latters first encode the image into the latent space and then invert the diffusion process (detailed in Table 3). Detection for these methods are performed in latent space, whereas post-hoc detectors operate directly on images.

---

> > > ### Comment · Reviewer_jcck · 2025-11-27
> > >
> > > Thank you for the author's response. I am glad to see that the author has clarified some of my confusion. To be more specific, my request about the system overview is not just about explaining how to train your Guidance method, but also include how your method is later used in watermarking systems. For example, you responded to my last question about how detection is performed, which should be included in the system figure, and should be compared against other methods (TR, GS) to show your advantage in a visualized way.

---

> > > > ### Author Response · Authors · 2025-11-27
> > > >
> > > > Thank you for your response. We are glad to see that our revisions have clarified most of the points you raised. Following your suggestion, we have updated the diagram to include the detection component of our guidance methods. We also provide a more complete figure, summarizing our guidance method, post-hoc methods and seed-based methods both at embedding and detection time. However due to space constraints this figure has been placed in Appendix E -- see Figure 21 in the revised paper.

---

### Official Review · Reviewer_WTJb · 2025-10-25

**Soundness:** 2
**Presentation:** 2
**Contribution:** 3
**Rating:** 6
**Confidence:** 3

**Summary:**

This paper proposes a new image watermarking method that enables converting an existing post-processing watermarking method into an in-generation watermarking method using only an existing decoder, without introducing significant changes to the watermarked image.

**Strengths:**

1.	Compared to existing post-generation and in-generation watermarking methods, the proposed approach offers improvements in watermark bit capacity.
2.	From the methodological point of view, the paper provides a practical means of converting post-processing watermarking approaches into in-generation ones.

**Weaknesses:**

1.	It’s unclear what motivates the authors to propose this method. The writing of the introduction should be revised to explicitly show this.
2.	It is unclear whether the method performs semantic watermarking or adds fixed content-unaware perturbations, as similar in-generation watermarking methods like Tree-Ring and Gaussian Shading have been found out to be adding fixed perturbations [1]. Since the proposed method is also an in-generation watermarking method, the authors should use a fixed watermarking key and test under [1]’s removal method to show whether or not the proposed watermark is content-aware.

[1] Yang et al., “Can simple averaging defeat modern watermarks,” in NeurIPS 2024.

**Questions:**

1.	For bold results presented in Table 1, would it be possible to also report the corresponding watermark decoding bit accuracy values? This could align the metric with most existing works and facilitate comparison.

---

> ### Author Response · Authors · 2025-11-20
>
> We thank the reviewer for their thorough evaluation. We appreciate their recognition of our method to convert post-hoc watermarking schemes into in-generation.
>
> ## Weaknesses
>
> > It’s unclear what motivates the authors to propose this method. The writing of the introduction should be revised to explicitly show this.
>
> Thank you for the comment. We clarify this in the paper in Section 3 *Motivation*. Our goal is to embed the watermark directly during the diffusion process, which, to the best of our knowledge, no other method achieves. Other in-generation methods are either seed-based (Tree-Rings, Gaussian-Shading) or VAE-based (Stable-Signature). Our approach additionally improves robustness against standard decoder augmentations and does not require any retraining.
>
> > It is unclear whether the method performs semantic watermarking or adds fixed content-unaware perturbations, as similar in-generation watermarking methods like Tree-Ring and Gaussian Shading have been found out to be adding fixed perturbations [1]. Since the proposed method is also an in-generation watermarking method, the authors should use a fixed watermarking key and test under [1]’s removal method to show whether or not the proposed watermark is content-aware.
>
> Thank you for this comment and for the reference. We added the results against  this attack as well as other adversarial attacks in Section 5.3 *Adversarial Attacks*. We also added a study of the dependency on the content compared to seed-based watermarking schemes in Appendix D.3.
> In our setup, we prefer using different keys to ensure security. Nevertheless, it is indeed interesting to explore how content-dependent the method is. We use the attack in its most powerful setting -- i.e. the grey-box version. We report in Figure 4 the bit accuracy (Gaussian-Shading) or $-\log_{10}P_{FA}@P_D=90\%$ (Tree-Rings) as a function of the PSNR between the attacked image and the watermarked image.
>
> We summarize herein the results in a table. **Our guidance method, G-VS, is the only one robust against this attack.**
> | **Attack** | **WM_scale** | PSNR (30, 32) | (32, 34) | (34, 36) | (36, 38) | (38, 40) |
> |---|---|---|---|---|---|---|
> | Average | Tree-Rings | 0.5 | 0.5 | 0.4 | 0.4 | 0.4 |
> |  | G-VS | **42.2** | **43.8** | **48.1** | **60.1** | **67.6** |
>
> | **Attack** | **WM_scale** | PSNR (30, 32) | (32, 34) | (34, 36) | (36, 38) | (38, 40) | (40, 42) |
> |---|---|---|---|---|---|---|---|
> | Average | Gaussian-Shading | 0.50 | 0.50 | 0.49 | 0.49 | 0.50 | -- |
> |  | G-VS | **0.82** | **0.84** | **0.86** | **0.89** | **0.91** | -- |
>
>
> ## Questions
>
> > For bold results presented in Table 1, would it be possible to also report the corresponding watermark decoding bit accuracy values? This could align the metric with most existing works and facilitate comparison.
>
> We chose to report capacity rather than bit accuracy, as it allows for a more meaningful comparison of different watermarking schemes in terms of the number of bits embedded. Note that it is directly related to bit accuracy. At this point, the table is already quite crowded with different metrics.
> Nevertheless, if the reviewer believes that the table would be more relevant with bit accuracy, we are open to discussion.
>
> In any case, we already report the bit accuracy in the original paper in Figure 13 in the Appendix D.
> Please find here a summary of the bit accuracy:
>
> | **LDM**  **WM** | Bit Accuracy |
> |---|---|
> | SD2 G-SSig | **0.99 (+0.12)** |
> | SD2 G-VS | **0.99 (+0.03)** |
> |---|---|
> | Flux G-SSig | **0.99 (+0.12)** |
> | Flux G-VS | **0.98 (+0.02)** |
> |---|---|
> | Sana G-SSig | **0.99 (+0.12)** |
> | Sana G-VS | **0.99 (+0.03)** |
> |---|---|

---

> ### Author Response · Authors · 2025-11-27
>
> Dear Reviewer WTjb,
>
> Thank you once again for your valuable feedback. We have conducted additional experiments and made revisions to the paper based on your suggestions. As the discussion phase is nearing its conclusion, we would like to know if our responses have addressed your concerns. We look forward to hearing from you.
>
> Best, Authors

---

### Official Review · Reviewer_wPqL · 2025-11-02

**Soundness:** 3
**Presentation:** 3
**Contribution:** 3
**Rating:** 6
**Confidence:** 4

**Summary:**

This study introduces a technique utilizing guidance to incorporate watermarks into the diffusion-based image creation workflow. By sourcing the guidance from an existing watermark decoder, it adapts after-the-fact watermarking approaches into on-the-fly methods that work seamlessly with various guided generation systems. The evaluations assert that it achieves durable watermarking while preserving the visual fidelity and meaning of the images.

**Strengths:**

1. Employing guidance strategies to enable in-generation watermark integration within diffusion frameworks opens up a compelling avenue for exploration.
2. The system's capacity to merge established after-the-fact watermarking methods with guided  generation processes offers significant utility in application.
3. Overall, the manuscript is structured effectively and easy to follow.

**Weaknesses:**

1.	For embedding watermarks in individual images, the approach imposes an excessively high computational burden. Conducting complete denoising operations to derive guidance at every noise level far surpasses the time required for standard generation (as evidenced in Appendix E). This greatly hampers its applicability to extensive image production scenarios.
2.	The tuning complexity prevents the method from being a true 'plug-and-play' solution. As illustrated in Appendix B, Figure 6, setting the guidance strength ($\omega$) too high can introduce severe visual artifacts into the generated image, necessitating careful calibration for different models and decoders.

**Questions:**

1.	The proposed method relies on a public, off-the-shelf decoder to compute gradients. Does this imply that an attacker, upon obtaining a watermarked image, could actively erase the watermark by computing and applying an 'adversarial' or 'reverse' guidance gradient aimed at minimizing the decoder's output (i.e., a white-box adaptive attack)?
2.	Regarding the 'fast guidance' experiments in Section 5.5, were the results (e.g., "G-VS last 15") obtained by applying *both* the 'step reduction' (simplification 1) and the 'gradient approximation' ($\nabla_{z_{0}}$ instead of $\nabla_{z_{t}}$, simplification 2) strategies simultaneously? If so, what is the respective contribution of each simplification strategy to the observed performance degradation?

---

> ### Author Response · Authors · 2025-11-20
> **Part 1/2**
>
> We thank the reviewer for their thoughtful review. We appreciate them  stressing the originality of the guided diffusion applied to watermarking.
>
> ## Weaknesses
>
> > For embedding watermarks in individual images, the approach imposes an excessively high computational burden. Conducting complete denoising operations to derive guidance at every noise level far surpasses the time required for standard generation (as evidenced in Appendix E). This greatly hampers its applicability to extensive image production scenarios.
>
> We acknowledge this as a limitation of the paper. However, to the best of our knowledge, our method is the only one that embeds the watermark directly during the diffusion process. As such, we believe that sutdying the feasability of this approach is an interesting research question by itself. We demonstrate in Table 3 that our method outperforms other in-gen approaches even when guidance is applied only on the last 10 steps. While our generation requires 45 additionnal steps, detection is  "instantaneous", whereas seed-based methods require additional diffusion steps for each detection.
>
> > The tuning complexity prevents the method from being a true 'plug-and-play' solution. As illustrated in Appendix B, Figure 6, setting the guidance strength () too high can introduce severe visual artifacts into the generated image, necessitating careful calibration for different models and decoders.
>
> Thank you for the remark. The calibration requires only a few images and generalizes well across a given model–detector pair. We do not claim that our choice of hyperparameters is optimal. We added an ablation study on the choice of hyperparameters in Appendix B.1. We show that the guidance watermarking scale controls the trade-off between quality and detectability. On the other hand, the clipping prevents extreme values. Such values are susceptible to create artifacts, such as those observable in Figure 8. Here is a table asked by reviewer 7E16. It gives clearer intuitions of the impact of each parameter. We advise the reviewer to refer to Figure 9 for better readability.
>
> | **Clip %** | **WM Scale** | PSNR | LPIPS | CLIP | Bit Accuracy |
> |---|---|---|---|---|---|
> | 0% | 0 | Inf | 0.0 | 0.344 | 0.500 |
> |  | 150 | 29.0 | 0.03 | 0.344 | 0.975 |
> |  | 300 | 25.9 | 0.05 | 0.345 | 0.989 |
> |  | 600 | 22.2 | 0.10 | 0.344 | 0.993 |
> |  | 900 | 19.5 | 0.18 | 0.342 | 0.996 |
> |  | 1200 | 17.6 | 0.24 | 0.340 | 0.997 |
> |---|---|---|---|---|
> | 10% | 0 | Inf | 0.0 | 0.344 | 0.500 |
> |  | 150 | 29.9 | 0.02 | 0.344 | 0.969 |
> |  | 300 | 27.5 | 0.04 | 0.344 | 0.987 |
> |  | 600 | 24.0 | 0.07 | 0.345 | 0.992 |
> |  | 900 | 21.5 | 0.12 | 0.345 | 0.995 |
> |  | 1200 | 20.1 | 0.16 | 0.342 | 0.996 |
> |---|---|---|---|---|
> | 0% | 600 | 22.2 | 0.10 | 0.344 | 0.993 |
> | 5% |  | 23.3 | 0.08 | 0.344 | 0.994 |
> | 10% |  | 24.0 | 0.07 | 0.345 | 0.992 |
> | 15% |  | 24.8 | 0.06 | 0.345 | 0.993 |
> | 20% |  | 25.1 | 0.06 | 0.345 | 0.992 |

---

> > ### Author Response · Authors · 2025-11-20
> > **Part 2/2**
> >
> > ## Questions
> >
> > > The proposed method relies on a public, off-the-shelf decoder to compute gradients. Does this imply that an attacker, upon obtaining a watermarked image, could actively erase the watermark by computing and applying an 'adversarial' or 'reverse' guidance gradient aimed at minimizing the decoder's output (i.e., a white-box adaptive attack)?
> >
> > Thank you for the question. We used publicly available off-the-shelf detectors for experimental and reproducibility purposes. However, we agree that for security reasons, employing a private detector would be preferable. As with any watermarking scheme, using a public detector makes it theoretically possible to perform adversarial attacks. Notably, this is the case of any of the current SOTA post-hoc watermark such as VideoSeal and is not, as such, a specific weakness of our work.
> >
> > > Regarding the 'fast guidance' experiments in Section 5.5, were the results (e.g., "G-VS last 15") obtained by applying both the 'step reduction' (simplification 1) and the 'gradient approximation' ( instead of , simplification 2) strategies simultaneously? If so, what is the respective contribution of each simplification strategy to the observed performance degradation?
> >
> > Thank you for your question, we added an ablation study in the Appendix B.2 on the impact of the simplifications to complete the study. To clarify, all experiments were conducted using *simplification 2* in order to make our approach computationnaly tractable. Interestingly, **this is at no cost to the performance of our method**, to the contrary.
> > To support this claim, find here an additional experiment to highlight the impact of *simplification 2*. We see that both quality and detectability improve when using it *simplification 2*. We hypothesize that, since the guidance makes the watermark signal content-dependent (see new Appendix D.3), the signal added during the first steps is erased by bigger semantic changes. On the other hand, *simplification 2* makes the signal depend more on local content, added at the end of the diffusion process. Consequently, the signal is more likely to be preserved during the diffusion process.
> >
> > *Simplification 1* has a detrimentral impact on the detectability the more steps we skip, as shown in the Table 3. However, less steps also translates to a large gain in computationnal cost as well as an improvements in image quality -- see the quality metrics in the table.
> > | **WM scheme** | PSNR | LPIPS | CLIP |
> > |---|---|---|---|
> > | G-VS | 23.5 | 0.07 | 0.348 |
> > | G-VS last 15 | 28.3 | 0.02 | 0.348 |
> > | G-VS last 10 | 30.6 | 0.01 | 0.347 |
> > | G-VS last 5 | 35.8 | 0.004 | 0.348 |
> >
> > | **WM scheme** | PSNR | LPIPS | CLIP | Capacity | $-\log_10 P_{FA}@PD=90%$ |
> > |---|---|---|---|---|---|
> > | G-VS (w/o simplification 2) | 8.8 | 0.67 | 0.17 | 140.7 | 49.1 |

---

> ### Author Response · Authors · 2025-11-27
>
> Dear Reviewer wPqL,
>
> Thank you once again for your valuable feedback. We have conducted additional experiments and made revisions to the paper based on your suggestions. As the discussion phase is nearing its conclusion, we would like to know if our responses have addressed your concerns. We look forward to hearing from you.
>
> Best, Authors

---

### Official Review · Reviewer_7E16 · 2025-11-11

**Soundness:** 3
**Presentation:** 3
**Contribution:** 2
**Rating:** 4
**Confidence:** 4

**Summary:**

The paper introduces a method that embeds digital watermarks directly during the diffusion sampling process. The approach leverages gradients from pretrained watermark decoder to guide the diffusion model toward generating images that the decoder recognizes as watermarked. It embeds the watermark during generation using a cosine-based loss that aligns decoder features with a target watermark vector, and aggregates gradients across image augmentations to enhance robustness to unseen attacks. No need to retrain the decoder.

**Strengths:**

1. The method is general and retraining-free. It can transform any existing post-hoc watermark decoder into an in-generation embedding scheme without retraining either the decoder or the diffusion model.
2. The method is empirically validated on multiple diffusion models and decoders.
3. The method has a unified formulation for multi-bit and zero-bit watermarking.

**Weaknesses:**

1. The approach requires a pretrained differentiable watermark decoder (e.g. stable signature). Its performance and the statical guarantees are bound by the robustness and isotropy of that decoder's feature space. If the decoder is biased or unavailable, the framework can't work.
2. The attack scope is narrow. The experiments focus on standard image augmentations (JPEG, crop, brightness). Other relevant attacks like regeneration attacks, adversarial attacks should also be tested.
3. No ablation on guidance scheduling. The method mentions turning on guidance only for the last diffusion steps (section 4.5). But there is little analysis on how the choice of step $T_w$, clipping thresholds, or gradient aggregation method affects detectability or quality.

**Questions:**

1. All experiments are conducted on latent diffusion model. I'm wondering its effectiveness  on pixel-based diffusion models.

---

> ### Author Response · Authors · 2025-11-20
> **Part 1/2**
>
> We thank the reviewer for the feedback. We appreciate their recognition that our method provides an effective approach for converting post-processing techniques into in-generation watermarking.
>
> ## Weaknesses
>
> > The approach requires a pretrained differentiable watermark decoder (e.g. stable signature). Its performance and the statical guarantees are bound by the robustness and isotropy of that decoder's feature space. If the decoder is biased or unavailable, the framework can't work.
>
> We already demonstrate that our method is not limited by the robustness of the detector -- see Figure 6 and Section 5.4 *Unknown Augmentations*. In our study, we assume the availability of an off-the-shelf detector. Naturally, without any detector, the guidance cannot be performed, just as detecting Gaussian-Shading or Tree-Rings assumes access the correspondive diffusion model.
> The isotropy of the detector is not required for using our guidance. It is only required if the user wants to optimize a statistically sound $p$-value. Furthermore, we provide a complete yet simple methodology in Appendix C to convert a biased detector into a detector usable within our statistical framework. These whitened versions of the detectors where used succesfully in the paper, ensuring statistical guarantees and fair comparisons between different watermarking schemes, showing the validity and generality of this approach.
>
> > The attack scope is narrow. The experiments focus on standard image augmentations (JPEG, crop, brightness). Other relevant attacks like regeneration attacks, adversarial attacks should also be tested.
>
> Thank you for your remark. Even though we performed VAE purification attacks in the original experiments (see Section 5.4), we agree that the scope should be enlarged.
> As such, we added a study about adversarial attacks with, VAE attack, grey-box average attack and Regen attack in Section 5.3 *Adversarial attacks*. Here are tables of the results that summarize the results. We advise the reviewer to refer to Figures 4 in the paper for better readability. The first table gives the mean $-log_{10}(p_{FA})@P_D=90\%$ for the corresponding PSNR bin for Tree-Rings and G-VS. The second one gives the mean bit accuracy for the corresponding PSNR bin for Gaussian-Shading and G-VS.
> | **Attack** | **WM_scale** | PSNR (30, 32) | (32, 34) | (34, 36) | (36, 38) | (38, 40) |
> |---|---|---|---|---|---|---|
> | VAE purification | Tree-Rings | 15.8 | 19.4 | 9.7 | 14.6 | 12.7 |
> |  | G-VS | **45.1** | **50.6** | **48.4** | **48.3** | -- |
> | Regeneration | Tree-Rings | 13.5 | 11.8 | 8.8 | 13.3 | -- |
> |  | G-VS | **31.2** | **41.6** | **47.1** | **31.7** | -- |
> | Average | Tree-Rings | 0.3 | 0.3 | 0.3 | 0.2 | 0.2 |
> |  | G-VS | **13.2** | **21.8** | **21.6** | **29.5** | **37.4** |
>
> | **Attack** | **WM_scale** | PSNR (30, 32) | (32, 34) | (34, 36) | (36, 38) | (38, 40) | (40, 42) |
> |---|---|---|---|---|---|---|---|
> | VAE purification | Gaussian-Shading | **0.97** | **0.96** | **0.96** | **0.94** | **0.90** | **0.83** |
> |  | G-VS | 0.91 | 0.92 | 0.93 | 0.89 | -- | -- |
> | Regeneration | Gaussian-Shading | **0.95** | **0.92** | **0.94** | **0.89** | **0.85** | -- |
> |  | G-VS | 0.88 | 0.89 | 0.91 | 0.87 | -- | -- |
> | Average | Gaussian-Shading | 0.50 | 0.50 | 0.49 | 0.49 | 0.50 | -- |
> |  | G-VS | **0.82** | **0.84** | **0.86** | **0.89** | **0.91** | -- |
>
> G-VS is always more robust against these attacks than Tree-Rings. But Gaussian-Shading, due to the redundancy of the signal is more robust than G-VS against VAE purification and Regeneration. Note that G-VS achieve quite high bit accuracies even if they are lower than Gaussian-Shading for these attacks. The biggest advantage of G-VS compared to seed-based methods is that it is content-dependent as explained in Appendix D.3. This is not the case of Gaussian-Shading and Tree-Rings, making **the average attack very effective against seed-based scheme whereas G-VS is robust**.

---

> > ### Author Response · Authors · 2025-11-20
> > **Part 2/2**
> >
> > > No ablation on guidance scheduling. The method mentions turning on guidance only for the last diffusion steps (section 4.5). But there is little analysis on how the choice of step, clipping thresholds, or gradient aggregation method affects detectability or quality.
> >
> > Thank you for the remark. We do not claim that our choice of the hyperparameters is optimal. Empirically, we observed that varying them has limited impact on both detection and image quality. The guidance parameter controls the watermark strength. Without gradient clipping, stronger guidance may degrade visual quality and introduce artifacts.
> > We added an ablation on the guidance scale and the clipping in the Appendix B. We plot the bit accuracy and the PSNR as a function of the wm scale (see Figure 9). We also plot the bit accuracy and the PSNR as a function of the clip %.
> >
> > Here is a table with the informations for Sana G-VS. Once again, we advise the reviewer to refer to Figure 9 within the paper itself.
> >
> > | **Clip %** | **WM Scale** | PSNR | LPIPS | CLIP | Bit Accuracy |
> > |---|---|---|---|---|---|
> > | 0% | 0 | Inf | 0.0 | 0.344 | 0.500 |
> > |  | 150 | 29.0 | 0.03 | 0.344 | 0.975 |
> > |  | 300 | 25.9 | 0.05 | 0.345 | 0.989 |
> > |  | 600 | 22.2 | 0.10 | 0.344 | 0.993 |
> > |  | 900 | 19.5 | 0.18 | 0.342 | 0.996 |
> > |  | 1200 | 17.6 | 0.24 | 0.340 | 0.997 |
> > |---|---|---|---|---|
> > | 10% | 0 | Inf | 0.0 | 0.344 | 0.500 |
> > |  | 150 | 29.9 | 0.02 | 0.344 | 0.969 |
> > |  | 300 | 27.5 | 0.04 | 0.344 | 0.987 |
> > |  | 600 | 24.0 | 0.07 | 0.345 | 0.992 |
> > |  | 900 | 21.5 | 0.12 | 0.345 | 0.995 |
> > |  | 1200 | 20.1 | 0.16 | 0.342 | 0.996 |
> > |---|---|---|---|---|
> > | 0% | 600 | 22.2 | 0.10 | 0.344 | 0.993 |
> > | 5% |  | 23.3 | 0.08 | 0.344 | 0.994 |
> > | 10% |  | 24.0 | 0.07 | 0.345 | 0.992 |
> > | 15% |  | 24.8 | 0.06 | 0.345 | 0.993 |
> > | 20% |  | 25.1 | 0.06 | 0.345 | 0.992 |
> >
> > Increasing the watermarking guidance scale improves bit accuracy but also amplifies the deviation from the non-watermarked image, as reflected by a lower PSNR. The parameter clip % prevents the apparition of extreme values, limiting visual artifacts. Increasing clip % results in a higher PSNR with only a minor reduction in bit accuracy.
> >
> >
> > ## Questions
> >
> > > All experiments are conducted on latent diffusion model. I'm wondering its effectiveness on pixel-based diffusion models.
> >
> > It is true that we did not include experiments on pixel-based diffusion models, as our focus was on latent-based architectures which is the current the state-of-the-art for image generation. The use of pure Diffusion Denoising Probabilistic Models (DDPM) in the pixel-domain has fallen out of favor due to its lesser quality and higher computationnal cost. As such, we don't see the necessity of adding such experiment in the main body of the paper. Nevertheless, since the latent space can be seen as a compact pixel-level representation, we expect that our method would perform similarly on pixel-based models.

---

> ### Author Response · Authors · 2025-11-27
>
> Dear Reviewer 7E16,
>
> Thank you once again for your valuable feedback. We have conducted additional experiments and made revisions to the paper based on your suggestions. As the discussion phase is nearing its conclusion, we would like to know if our responses have addressed your concerns. We look forward to hearing from you.
>
> Best, Authors

---

### Author Response · Authors · 2025-11-25
**Summary of modifications and improvements asked by reviewers**

We thank the reviewers for their thoughtful comments, which have helped us to significantly improve the paper.

We have made several modifications in response to the reviewers’ remarks:
- We enhanced our comparison with In-Gen methods by adding two additional augmentations (Gaussian blur and rotation) to Table 2 -- *as asked by reviewer jcck*. **Our guidance remains consistently better detected than both Gaussian-Shading and Tree-Rings**.
- We also included a paragraph in Section 5.3 (Adversarial Attacks) comparing our method to In-Gen against relevant attacks: VAE purification, regeneration and averaging attacks -- *as asked by reviewers 7E16 and WTjb*. Our guidance is always more robust than Tree-Rings, especially under the average attack. We also provide empirical evidence that Gaussian-Shading is slightly more robust to regeneration and VAE purification due to redundancy of its signal, but it completely fails under the average attack, whereas **our guidance remains robust against all three attacks**.
- **Appendix B has been updated with two ablation studies, highlighting the impact of hyperparameters and each simplification** (Section 4.5 *Fast and controlled guidance for watermarking*, Section 5.5 *Computation cost*) -- *as requested by reviewers 7E16, wPqL and jcck*.
- We added an ablation study on content-dependency in Appendix D.3 (Dependency on the Content) to compare our method with classic In-Gen approaches -- as asked by reviewer WTJb. **We demonstrate that our method is content-dependent**, whereas seed-based methods are not. **This makes our guidance method more secure than other in-gen approaches**.
- System-level diagrams and the Appendix E *Methodology comparison with other Watermarking schemes* have been added to provide a more comprehensive and visual comparison between watermarking approaches -- *as asked by reviewer jcck* -- see Figures 1 and 21.

---

### Meta-Review · Area_Chair_sr6B · 2025-12-26

**Summary:**

This paper received initial review ratings of 4, 6, 6 and 4. The reviewers appreciated the training-free design, the evaluation on multiple diffusion models and decoders, the well-structured manuscript, and the employment of guidance strategies. They raised questions, including unclear motivation, adversarial attacks, parameters setting, and complexity. During the rebuttal, the authors provided detailed responses with additional experiments. Before the stop of the rebuttal discussion, only reviewer Jcck replied to the authors and agreed that the authors clarified some of their confusion. The AC read the reviewers’ comments and the reviewers’ rebuttal in detail. He believes that the authors answered most of the reviewer questions, but he has concerns about the high computational cost, which may limit the useability of the proposed method. The authors acknowledged this limit. Taking into account the strengths and weaknesses of the proposed method and overall score distribution of ICLR 2026, the AC recommends accepting this paper with reserve.

**Reviewer Concerns:**

The authors answered almost all the questions from the reviewers. They acknowledge that computation cost is the current limit.

**Reviewer Scores:**

I think that 7E16, whose initial score was 4, may increase to 6. However, I feel that other reviewers may not change their scores, in particular wPqL and WTJb, whose initial scores are 6.

---

### Decision · Program_Chairs · 2026-01-26

Accept (Poster)